# Reinforced Few-Shot Acquisition Function Learning for Bayesian Optimization

**Bing-Jing Hsieh[1], Ping-Chun Hsieh[1], Xi Liu[2]**
[1]Department of Computer Science, National Yang Ming Chiao Tung University, Hsinchu, Taiwan
[2]Applied Machine Learning, Facebook AI, Menlo Park, CA, USA
{bingjing2000.cs08g, pinghsieh}@nctu.edu.tw, xliu1@fb.com

## Abstract

Bayesian optimization (BO) conventionally relies on handcrafted acquisition functions (AFs) to sequentially determine the sample points. However, it has been widely observed in practice that the best-performing AF in terms of regret can vary significantly under different types of black-box functions. It has remained a challenge to design one AF that can attain the best performance over a wide variety of black-box functions. This paper aims to attack this challenge through the perspective of reinforced few-shot AF learning (FSAF). Specifically, we first connect the notion of AFs with Q-functions and view a deep Q-network (DQN) as a surrogate differentiable AF. While it serves as a natural idea to combine DQN and an existing few-shot learning method, we identify that such a direct combination does not perform well due to severe overfitting, which is particularly critical in BO due to the need of a versatile sampling policy. To address this, we present a Bayesian variant of DQN with the following three features: (i) It learns a distribution of Q-networks as AFs based on the Kullback-Leibler regularization framework. This inherently provides the uncertainty required in sampling for BO and mitigates overfitting. (ii) For the prior of the Bayesian DQN, we propose to use a demo policy induced by an off-the-shelf AF for better training stability. (iii) On the meta-level, we leverage the meta-loss of Bayesian model-agnostic meta-learning, which serves as a natural companion to the proposed FSAF. Moreover, with the proper design of the Q-networks, FSAF is general-purpose in that it is agnostic to the dimension and the cardinality of the input domain. Through extensive experiments, we demonstrate that the FSAF achieves comparable or better regrets than the state-of-the-art benchmarks on a wide variety of synthetic and real-world test functions.

## 1 Introduction

Bayesian optimization (BO) has served as a powerful and popular framework for global optimization in many real-world tasks, such as hyperparameter tuning [1–4], robot control [5], automatic material design [6–8], etc. To search for global optima under a small sampling budget and potentially noisy observations, BO imposes a Gaussian process (GP) prior on the unknown black-box function and continually updates the posterior as more samples are collected. BO relies on *acquisition functions* (AFs) to determine the sample location, i.e., those with larger AF values are prioritized than those with smaller ones. AFs are often designed to capture the trade-off between exploration and exploitation of the global optima. The design of AFs has been extensively studied from various perspectives, such as optimism in the face of uncertainty (e.g., GP-UCB [9]), optimizing information-theoretic metrics (e.g., entropy search methods [10–12]), and maximizing one-step improvement (e.g., expected improvement or EI [13, 14]). As a result, AFs are often handcrafted according to different perspectives of the trade-off, and the best-performing AFs can vary significantly under different types of black-box

35th Conference on Neural Information Processing Systems (NeurIPS 2021).

functions [11]. This phenomenon is repeatedly observed in our experiments in Section 4. Therefore, one critical issue in BO is to design an AF that can adapt to a variety of black-box functions.

To achieve better adaptability to new tasks in BO, recent works propose to leverage *meta-data*, the data previously collected from similar tasks [15–17]. For example, in the context of hyperparameter optimization under a specific dataset, the meta-data could come from the evaluation of previous hyperparameter configurations for the same learning model over any other related dataset. In [15], meta-data is used to fine-tune the initialization of the GP parameters and thereby achieves better GP model selection for each specific task. However, the potential benefit of using meta-data for more efficient exploration via AFs is not explored. On the other hand, in [16, 17], meta-data is split into multiple subsets and then used to construct a transferable acquisition function based on some off-the-shelf AF (e.g. EI) and an ensemble of GP models. Each of the GP models is learned over a separate subset of the meta-data. However, to achieve effective knowledge transfer, this approach would require a sufficiently large amount of meta-data, which significantly limits its practical use. As a result, there remains a critical unexplored challenge in BO: *how to design an AF that can effectively adapt to a wide variety of back-box functions given only a small amount of meta-data?*

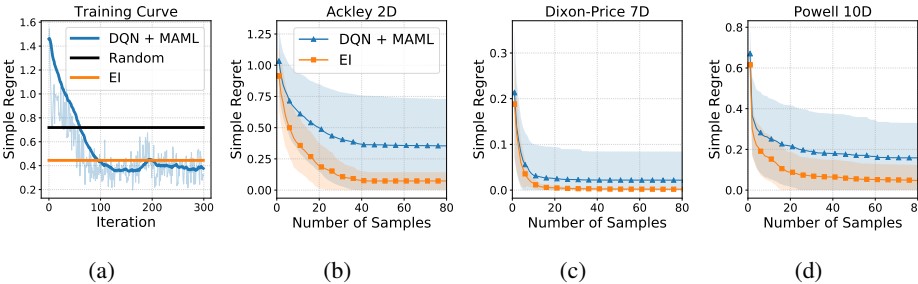

Figure 1: An illustration of the overfitting issue of DQN+MAML trained with GP functions: (a) Training curves of DQN+MAML (As EI and Random do not require any training, their lines are flat); (b)-(d) Average simple regrets of DQN+MAML and EI in testing under benchmark functions.

To tackle this challenge, we propose to rethink the use of meta-data in BO through the lens of *few-shot acquisition function (FSAF) learning*. Specifically, our goal is to learn an initial AF model that allows few-shot fast adaptation to each specific task during evaluation. Inspired by the similarity between AFs and Q-functions, we use a deep Q-network (DQN) as a surrogate differentiable AF, i.e., the Q-network would output an indicator for each candidate sample point given its posterior mean and variance as well as other related information. Given the parametric nature and the differentiability of a Q-network, it is natural to leverage optimization-based few-shot learning approaches, such as model-agnostic meta-learning (MAML) [18], for the training of few-shot AFs. Despite this natural connection, we find that a direct combination of standard DQN and MAML (DQN+MAML) is prone to overfitting, as illustrated by the comparison of training and testing results. Note that in Figure 1 (with detailed configuration in Appendix C), DQN+MAML achieves better regret than EI on the training set but suffers from much higher regret during testing. We hypothesize this is because both DQN and MAML are prone to overfitting [19–22]. This issue could be particularly critical in BO due to the uncertainty required in adaptive sampling. Based on our findings and inspired by [23], we propose a Bayesian variant of DQN with the following three salient features: (i) The Bayesian DQN learns a distribution of parameters of DQN based on the Kullback-Leibler (KL) regularization framework. Through this, the problem of minimizing the Q-learning temporal-difference (TD) error in DQN is converted into a Bayesian inference problem. (ii) For the prior of the Bayesian DQN, we propose to use a demonstration policy induced by an off-the-shelf AF to further stabilize the training process; (iii) We then use the chaser meta-loss in [20], which serves as a natural companion to the proposed Bayesian DQN. As shown by the experimental results in Section 4, the proposed design effectively mitigates overfitting and achieves good generalization under various black-box functions. Moreover, FSAF is trained solely on synthetic GP functions (with details in Appendix B) and is able to adapt to a broad variety of functions based only on a small amount of metadata (which could be costly to generate). With the proper design of the Q-networks, FSAF is general-purpose in the sense that it is agnostic to both the dimension and the cardinality of the input domain.

The main contributions of this paper can be summarized as follows:

- We consider a novel setting of few-shot acquisition function learning for BO and present the first few-shot acquisition function that can use a small amount of meta-data to achieve better task-specific exploration and thereby effectively adapt to a wide variety of black-box functions.
- Inspired by the similarity between AFs and Q-functions, we view DQN as a parametric and differentiable AF and use it as the base of our FSAF. We identify the important overfitting issue in the direct combination of DQN and MAML and thereafter present a Bayesian variant of DQN that mitigates overfitting and enjoys stable training through a demo-based prior.
- We extensively evaluate the proposed FSAF in a variety of tasks, including optimization benchmark functions, real-world datasets, and synthetic GP functions. We show that the proposed FSAF can indeed effectively adapt to a variety of tasks and outperform both the conventional benchmark AFs as well as the recent state-of-the-art meta-learning BO methods.

## 2 Preliminaries

Our goal is to design a sampling policy to optimize a black-box function $f : \mathbb{X} \to \mathbb{R}$, where $\mathbb{X} \subset \mathbb{R}^d$ denotes the compact domain of $f$. $f$ is black-box in the sense that there are no special structural properties (e.g., concavity or linearity) or derivatives (e.g., gradient or Hessian) about $f$ available to the sampling policy. In each step $t$, the policy selects $x_t \in \mathbb{X}$ and obtains a noisy observation $y_t = f(x_t) + \varepsilon_t$, where $\varepsilon_t$ are i.i.d. zero-mean Gaussian noises. To evaluate a sampling policy, we define the *simple regret* as $\text{Regret}(t) := \max_{x \in \mathbb{X}} f(x^*) - \max_{1 \le s \le t} f(x_s)$, which quantifies the best sample up to $t$. For convenience, we let $\mathcal{F}_t := \{(x_i, y_i)\}_{i=1}^{t-1}$ denote the observations up to step $t$.

**Bayesian optimization.** To optimize $f$ in a sample-efficient manner, BO first imposes on the space of objective functions a GP prior, which is fully characterized by a mean function and a covariance function, and then determines the next sample based on the resulting posterior [24, 25]. In each step $t$, given $\mathcal{F}_t$, the posterior predictive distribution of each $x \in \mathbb{X}$ is $\mathcal{N}(\mu_t(x), \sigma_t^2(x))$, where $\mu_t(x) := \mathbb{E}[f(x) | \mathcal{F}_t]$ and $\sigma_t(x) := (\mathbb{V}[f(x) | \mathcal{F}_t])^{\frac{1}{2}}$ can be derived in closed form. In this way, BO can be viewed as a sequential decision making problem. However, it is typically difficult to obtain the exact optimal policy due to the curse of dimensionality [24]. To obtain tractable policies, BO algorithms construct AFs $\Psi(x; \mathcal{F}_t)$, which resort to maximizing one-step look-ahead objectives based on $\mathcal{F}_t$ and the posterior [24]. For example, EI chooses the sample location based on the improvement made by the immediate next sample in expectation, i.e., $\Psi_{\text{EI}}(x; \mathcal{F}_t) = \mathbb{E}[f(x) - \max_{1 \le i \le t-1} f(x_i) | \mathcal{F}_t]$, which enjoys a closed form in $\mu_t(x)$, $\sigma_t(x)$, and $\max_{1 \le i \le t-1} f(x_i)$.

**Meta-learning with few-shot fast adaptation.** Meta-learning is a generic paradigm for generalizing the knowledge acquired during training to solving unseen tasks in the testing phase. In the few-shot setting, meta-learning is typically achieved via a bi-level framework: (i) On the upper level, the training algorithm is meant to determine a proper initial model with an aim to facilitating subsequent task-specific adaptation; (ii) On the lower level, given the initial model and the task of interest, a fast adaptation subroutine is configured to fine-tune the initial model based on a small amount of task-specific data. Specifically, during training, the learner finds a model parameterized by $\theta$ based on a collection of tasks $\mathcal{T}$, where each task $\tau \in \mathcal{T}$ is associated with a training set $\mathcal{D}_\tau^{\text{tr}}$ and a validation set $\mathcal{D}_\tau^{\text{val}}$. For any initial model parameters $\theta$ and training set $\mathcal{D}^{\text{tr}}$ of a task $\tau$, let $\mathcal{M}(\theta, \mathcal{D}_\tau^{\text{tr}})$ be an algorithm that outputs the adapted model parameters by applying few-shot fast adaptation to $\theta$ based on $\mathcal{D}_\tau^{\text{tr}}$. The performance of the adapted model is evaluated on $\mathcal{D}_\tau^{\text{val}}$ by a meta-loss function $\mathcal{L}(\mathcal{M}(\theta, \mathcal{D}_\tau^{\text{tr}}), \mathcal{D}_\tau^{\text{val}})$. Accordingly, the overall training can be viewed as solving the following optimization problem:

$$\theta^* := \underset{\theta}{\text{argmin}} \sum_{\tau \in \mathcal{T}} \mathcal{L}(\mathcal{M}(\theta, \mathcal{D}_\tau^{\text{tr}}), \mathcal{D}_\tau^{\text{val}}). \tag{1}$$

By properly configuring the loss function, the formulation in (1) is readily applicable to various learning problems, including supervised learning and RL. Note that the adaptation subroutine is typically chosen as taking one or a few gradient steps with respect to $\mathcal{L}(\cdot, \cdot)$ or some relevant loss function. For example, under the celebrated MAML [18], the adaptation subroutine is $\mathcal{M}(\theta, \mathcal{D}_\tau^{\text{tr}}) \equiv \theta - \eta \nabla_\theta \mathcal{L}(\theta, \mathcal{D}_\tau^{\text{tr}})$, where $\eta$ denotes the learning rate.

**Reinforcement learning and Q-function.** Following the conventions of RL, we use $s_t$, $a_t$, and $r_t$ to denote the state, action, and reward obtained at each step $t$. Let $R$ and $\gamma$ be the reward function and the discount factor. The goal is to find a stationary randomized policy that maximizes the total expected discounted reward $\mathbb{E}[\sum_{t=0}^{\infty} \gamma^t r_t]$. To achieve this, given a policy $\pi$, a helper

function termed Q-function is defined as $Q^\pi(s, a) := \mathbb{E}[\sum_{t=0}^\infty \gamma^t R(s_t, a_t)|s_o = s, a_0 = a; \pi]$. The optimal Q-function can then be defined as $Q^*(s, a) := \max_\pi Q^\pi(s, a)$, for each state-action pair. One fundamental property of the optimal Q-function is the *Bellman optimality equation*, i.e., $Q^*(s, a) = \mathbb{E}[r_t + \gamma \max_{a'} Q^*(s_{t+1}, a')|s_t = a, a_t = a]$. Then, the Bellman optimality operator can be defined by $[\mathcal{B}Q](s, a) := R(s, a) + \gamma \mathbb{E}_{s'}[\max_{a' \in \mathcal{A}} Q(s', a')]$. It is known that $Q^*$ is the unique fixed point of $\mathcal{B}$. We leverage this fact to describe the proposed FSAF in Section 3.2.

## 3 Few-Shot Acquisition Function

### 3.1 Deep Q-Network as a Differentiable Parametric Acquisition Function

Based on the conceptual similarity between acquisition functions and Q-functions, in this section we present how to cast a deep Q-network as a parametric and differentiable instance of acquisition function. To begin with, we consider the Q-network architecture with state and action representations as the input, as typically adopted by Q-learning for large action spaces [26].

**State-action representation.** In BO, an action corresponds to choosing one location to sample from the input domain $\mathbb{X}$, and the state at each step $t$ can be fully captured by the collection of sampled points $\{(x_i, y_i)\}_{i=1}^{t-1}$. However, this raw state representation appears problematic as its dimension depends on the number of observed sample points. Inspired by the acquisition functions, we leverage the posterior mean and variance as the *joint state-action representation* for each candidate sample location. In addition, we include the best observation so far (defined as $y_t^* := \mathrm{argmax}_{1 \le i \le t-1} y_i$) and the ratio between the current timestamp and total sampling budget $T$, which reflects the sampling progress in BO. In summary, at each step $t$, the state-action representation of each $x \in \mathbb{X}$ is designed to be a 4-tuple $(\mu_t(x), \sigma_t(x), y_t^*, \frac{t}{T})$, which is agnostic to the dimension and cardinality of $\mathbb{X}$.

**Reward signal.** To reflect the sampling progress, we define the reward $r_t$ as a function of the simple regret, i.e., $r_t = g(\mathrm{Regret}(t))$, where $g : \mathbb{R}_+ \to \mathbb{R}$ is a strictly decreasing function. Practical examples include $g(z) = -z$ and $g(z) = -\log z$.

**Remark 1** One popular approach to construct a representation of fixed dimension is through embedding. From this viewpoint, the posterior mean and variance can be viewed as a natural embedding generated by GP inference in the context of BO. It is an interesting direction to extend the proposed design by constructing more general state and action representations via embedding techniques.

**Remark 2** The main contribution of FSAF is to present the first few-shot learning based acquisition function for BO, and it is worth noting that FSAF is not the first approach that addresses BO through the lens of RL as policy-based RL has been adopted for solving BO in MetaBO [27]. As both FSAF and MetaBO address BO through the lens of RL, they share some common choices for the representation (e.g., $\mu_t(x)$ and $\sigma_t(x)$). Despite the similarity in the representation and the fact that using a good state-action representation is important in RL, we found that a good representation itself does not guarantee good regret performance. As shown by the results in Figure 1, DQN+MAML uses the same state-action representation as FSAF, but it still suffers from severe overfitting and poor regret. Accordingly, in this paper, we show that the key to achieving low regret under various black-box functions is the appropriate use of a small amount of metadata through the design of few-shot adaptation in FSAF.

### 3.2 A Bayesian Perspective of Deep Q-Learning for Bayesian Optimization

One major challenge in designing an acquisition function for BO is to address the wide variability of black-box functions and accordingly achieve a favorable explore-exploit trade-off for each task. To address this by deep Q-learning as described in Section 3.1, instead of learning a single Q-network as in the standard DQN [28], we propose to learn a distribution of Q-network parameters from a Bayesian inference perspective to achieve more robust exploration and more stable training. Inspired by [23], we adapt the regularized minimization problem to Q-learning in order to connect Q-learning and Bayesian inference as follows. Let $C(\theta)$ be the cost function that depends on the model parameter $\theta$. Instead of finding a single model, the Bayesian approach finds a distribution $q(\theta)$ over $\theta$ that minimizes the cost function augmented with a KL-regularization penalty, i.e.,

$$\min_{q(\theta)} \left\{ \mathbb{E}_{\theta \sim q(\theta)}[C(\theta)] + \alpha D_{\mathrm{KL}}(q \parallel q_0) \right\}, \tag{2}$$

where $q_0$ denotes a prior distribution over $\theta$ and $\alpha$ is a weight factor of the penalty and $D_{\mathrm{KL}}(\cdot \parallel \cdot)$ denotes the Kullback-Leibler (KL) divergence between two distributions. Note that $q(\theta)$ essentially

induces a distribution over the sampling policies $\pi_\theta$, and accordingly $q_0$ can be interpreted as constructing a prior over the policies. By setting the derivative of the objective in (2) with respect to the measure induced by $q$ to be zero, one can verify that the optimal solution to (2) is

$$q^*(\theta) = \frac{1}{Z} \exp\left(\frac{-C(\theta)}{\alpha}\right) q_0(\theta), \tag{3}$$

where $Z$ is the normalizing factor. One immediate interpretation of (3) is that $q^*(\theta)$ can be viewed as the posterior distribution under the prior distribution $q_0(\theta)$ and the likelihood $\exp(-C(\theta)/\alpha)$. To adapt the KL-regularized minimization framework to value-based RL for BO, the proposed FSAF algorithm is built on the following design principles for $C(\theta)$ and $q_0(\theta)$:

- **Use mean-squared TD error as the cost function:** Recall from Section 2 that the optimal Q-function is the fixed point of the Bellman optimality backup operation. Hence, we have $\mathcal{B}Q = Q$ if and only if $Q$ is the optimal Q-function. Based on this observation, one principled choice of $C(\theta)$ is the squared TD error under the operator $\mathcal{B}$, i.e., $\|\mathcal{B}Q - Q\|_2^2$. Moreover, in practice, DQN typically incorporates a replay buffer $\mathcal{R}_Q$ (termed the Q-replay buffer) as well as a target Q-network to achieve better training stability [28]. Therefore, we choose the cost function as

$$C(\theta) = \mathbb{E}_{(s,a,s',r)\sim\rho}\left[\left(\left(r + \gamma \max_{a'\in\mathcal{A}} Q(s',a';\theta^-)\right) - Q(s,a;\theta)\right)^2\right], \tag{4}$$

where $\rho$ denotes the underlying sample distribution of the replay buffer and $\theta^-$ is the parameter of the target Q-network[1]. In practice, the cost $C(\theta)$ is estimated by the empirical average over a mini-batch $\mathcal{D}$ of samples $(s,a,r,s')$ drawn from the replay buffer, i.e., $C(\theta) \approx \hat{C}(\theta) = \frac{1}{|\mathcal{D}|}\sum_{(s,a,r,s')\in\mathcal{D}}((r + \gamma\max_{a'\in\mathcal{A}} Q(s',a';\theta^-)) - Q(s,a;\theta))^2$.

- **Construct an informative prior with the help of the existing acquisition functions:** In (2), the KL-penalty with respect to a prior distribution is meant to encode prior domain knowledge as well as provide regularization that prevents the learned parameter from collapsing into a point estimate. One commonly-used choice is a uniform prior (i.e., $q(\theta) = c$ for some positive constant $c$), under which the KL-penalty reduces to the negative entropy of $q$. Given that BO is designed to optimize expensive-to-evaluate functions, it is therefore preferred to use a more informative prior for better sample efficiency. Based on the above, we propose to construct a prior with the help of a demo policy $\pi_D$ induced by existing popular AFs (e.g., EI or PI), which inherently capture critical information structure of the GP posterior. Define a similarity indicator $\delta(\pi_\theta, \pi_D)$ of $\pi_\theta$ and $\pi_D$ as

$$\delta(\pi_\theta, \tilde{\pi}) := \mathbb{E}_{s\sim\rho, a\sim\pi_D(\cdot|s)}\left[\log(\pi_\theta(s,a))\right], \tag{5}$$

where we slightly abuse the notation and let $\rho$ denote the state distribution induced by the replay buffers. Since the term $\log(\pi_\theta(s,a))$ in (5) is the log-likelihood of that the action of $\pi_\theta$ matches that of $\pi_D$ at a state $s$, $\delta(\pi_\theta, \pi_D)$ reflects how similar the two policies are on average (with respect to the state visitation distribution of $\pi_\theta$). Accordingly, we propose to design the prior $q_0(\theta)$ to be

$$q_o(\theta) \propto \exp\left(\delta(\pi_\theta, \pi_D)\right). \tag{6}$$

As it is typically difficult to directly evaluate $\delta(\pi_\theta, \pi_D)$ in practice, we construct another replay buffer $\mathcal{R}_D$ (termed the *demo* replay buffer), which stores the state-action pairs produced under the state distribution $\rho$ and the demo policy $\pi_D$, and estimate $\delta(\pi_\theta, \pi_D)$ by the empirical average over a mini-batch $\mathcal{D}'$ of state-action pairs drawn from the demo replay buffer, i.e., $\delta(\pi_\theta, \pi_D) \approx \hat{\delta}(\pi_\theta, \pi_D) = \frac{1}{|\mathcal{D}'|}\sum_{(s,a)\in\mathcal{D}'}\log(\pi_\theta(s,a))$.

- **Update the Q-networks by Stein variational gradient descent:** The solution in (3) is typically intractable to evaluate and hence approximation is required. We leverage the Stein variational gradient descent (SVGD), which is a general-purpose approach for Bayesian inference. Specifically, we build up $N$ instances of Q-networks (also called *particles* in the context of the variational methods) and update the parameters via Bayesian inference. Let $\theta^{(n)}$ denote the parameters of the

---

[1]In (4), the cost function depends implicitly on the target network parameterized by $\theta^-$. Despite this, as the target network is updated periodically from $Q(\cdot, \cdot; \theta)$, for notational convenience we do not make explicit the dependence of the cost function on $\theta^-$ in the notation $C(\theta)$.

$n$-th Q-network and use $\Theta$ as a shorthand of $\{\theta^{(n)}\}_{n=1}^{N}$. Under SVGD [29] and the prior described in (6), the Stein variational gradient of each particle can be derived as

$$g^{(n)}(\Theta) = \frac{1}{N} \sum_{i=1}^{N} \nabla_{\theta^{(i)}} \left( \frac{-1}{\alpha} C(\theta^{(i)}) + \delta(\pi_{\theta^{(i)}}, \pi_{\mathrm{D}}) \right) k(\theta^{(i)}, \theta^{(n)}) + \nabla_{\theta^{(i)}} k(\theta^{(i)}, \theta^{(n)}), \quad (7)$$

where $k(\cdot, \cdot)$ is a kernel function. As mentioned above, in practice $C(\theta)$ and $\delta(\pi_\theta, \pi_{\mathrm{D}})$ are estimated by the corresponding empirical $\hat{C}(\theta)$ and $\hat{\delta}(\pi_\theta, \pi_{\mathrm{D}})$, respectively. Let $\hat{g}^{(n)}(\Theta)$ denote the estimated Stein variational gradient based on $\hat{C}(\theta)$ and $\hat{\delta}(\pi_\theta, \pi_{\mathrm{D}})$. Accordingly, the parameters of each Q-network are updated iteratively by SVGD as

$$\theta^{(n)} \leftarrow \theta^{(n)} + \eta \cdot \hat{g}^{(n)}(\Theta), \quad (8)$$

where $\eta$ is the learning rate. For ease of notation, we use $\mathcal{M}_{\mathrm{SVGD}}(\Theta, \mathcal{D})$ to denote the subroutine that applies one SVGD update of (8) to all Q-networks parameterized by $\Theta$ based on some dataset $\mathcal{D}$. Note that the update scheme in (8) serves as a natural candidate for the few-shot adaptation subroutine of the meta-learning framework described in Section 2. In Section 3.3, we will put everything together and describe the full meta-learning algorithm of FSAF.

**Remark 3** The presented Bayesian DQN bears some high-level resemblance to the prior works [30–33], which are inspired by the classic principle of Thompson sampling for exploration. In [31, 32], the Q-function is assumed to be linearly parameterized with a Gaussian prior on the parameters such that the posterior can be computed in closed form. On the other hand, without imposing the linearity assumption, [32] approximates the intractable posterior by maintaining an ensemble of Q-networks as a practical heuristic of Thompson sampling to DQN. Different from [30–32], we approach Bayesian DQN through the principled framework of KL regularization for parametric Bayesian inference and solve it via SVGD, without any linearity assumption. [33] starts from an entropy-regularized formulation for Q-learning and assumes that the target Q-value is drawn from a Gaussian model to obtain a tractable posterior from the perspective of variational inference. By contrast, we do not rely on the Gaussian assumption and directly find the posterior by SVGD, and for more efficient training we consider a prior induced by an acquisition function. More importantly, the presented Bayesian DQN naturally helps substantiate the few-shot learning framework described in Section 2 for BO.

**Remark 4** The regularized formulation in (2) has been extensively applied in the class of policy-based methods in RL. For example, the entropy-regularized policy optimization [34] has been applied to enable a "soft version" of policy iteration, which gives rise to the popular soft Q-learning [35] and soft actor-critic algorithms [36]. Another example is the Stein variational policy gradient method [23], which connects the policy gradient methods with Bayesian inference. Different from the prior works, we take a different path to connect the value-based RL approach with Bayesian inference.

**Remark 5** The similarity indicator defined in (5) has a similar form as the loss term in some of the classic imitation learning algorithms. For example, given an expert policy $\pi_e$, DAgger [37] is designed to find a policy $\pi'$ that minimizes a surrogate loss $\mathbb{E}_s[\ell(s, \pi_e)]$, where $\ell(\cdot, \cdot)$ is some loss function (e.g., 0-1 loss or hinge loss) that reflects the dissimilarity between $\pi'$ and $\pi_e$. Despite this high-level resemblance, one fundamental difference between (5) and imitation learning is that the demo policy $\pi_{\mathrm{D}}$ is not a true expert in the sense that mimicking the behavior of $\pi_{\mathrm{D}}$ is not the ultimate goal of the FSAF learning process. Instead, the goal of FSAF is to learn a policy that can better adapt to new tasks and thereby outperform the existing AFs in various domains. The penalty defined in (6) is only meant to provide some prior domain knowledge to achieve more sample-efficient training. We further validate this design by providing training curves in Section 4.

### 3.3 Meta-Learning via Bayesian MAML

Based on the Bayesian DQN design in Section 3.2, the natural way to substantiate the meta-learning framework in (1) is to leverage the Bayesian variant of MAML [20]. In the context of BO, each task typically corresponds to optimizing some type of black-box functions (e.g., GP functions from an RBF kernel with some lengthscale). FSAF implements the bi-level framework of (1) as follows: (i) On the lower level, for each task $\tau$, FSAF enforces few-shot fast adaptation by taking $K$ steps of SVGD as described in (8) and thereafter obtains the fast-adapted parameters denoted by $\Theta_{\tau,K}$; (ii) On the upper level, for each task $\tau$, FSAF computes a meta-loss that reflects the dissimilarity between the approximated posterior induced by $\Theta_{\tau,K}$ and the true posterior distribution in (6). As the

true posterior is not available, one practical solution is to approximate the true posterior by taking $S$ additional SVGD gradient steps based on $\Theta_{\tau,K}$ and obtaining a surrogate denoted by $\Theta^*_{\tau,S}$ [20]. This design can be justified by the fact that $\Theta^*_{\tau,S}$ becomes a better approximation for the true posterior as $S$ increases due to the nature of SVGD. For any two collections of particles $\Theta' \equiv \{\theta'^{(n)}\}$ and $\Theta'' \equiv \{\theta''^{(n)}\}$, define $D(\Theta', \Theta'') := \sum_{n=1}^{N} \|\theta'^{(n)} - \theta''^{(n)}\|_2^2$ (called *chaser loss* in [20]). Then, for any task $\tau$, the meta-loss of FSAF is computed as

$$\mathcal{L}_{\text{meta}}(\Theta; \tau) = D\big(\Theta_{\tau,K}, \text{stopgrad}(\Theta^*_{\tau,S})\big), \tag{9}$$

As will be shown by the experiments in Section 4, using small $K$ and $S$ empirically leads to favorable performance. The pseudo code of the training procedure of FSAF is provided in Appendix A.

**Remark 6** Based on the convention of few-shot learning for RL, one *shot* in few-shot adaptation refers to a trajectory or a rollout generated under the target task. For example, in a MuJoCo control task, a shot corresponds to a complete trajectory of states and actions collected under a policy. In the context of BO, *one-shot adaptation* means that each Q-network performs sequential sampling based on its Q-value for generating one trajectory and then uses this trajectory for the gradient updates for fast adaptation. The above notion is consistent with that in Bayesian MAML [20, Appendix C.2].

**Remark 7** This paper focuses on value-based methods for training a few-shot acquisition function. Based on the proposed training framework, it is also possible to extend the idea to design an actor-critic counterpart of our FSAF. We believe this is an interesting direction for future work.

## 4   Experimental Results

We demonstrate the effectiveness of FSAF on a wide variety of classes of black-box functions and discuss how FSAF addresses the critical challenges described in Section 1. Unless stated otherwise, we report the mean simple regrets over 100 evaluation trials.

**Popular benchmark methods.** We evaluate FSAF against various popular benchmark methods, including EI [13], PI [38], GP-UCB [9], MES [12], TAF [16], Spearmint [1], HEBO [39], and MetaBO [27]. The configuration and hyperparameters of the above methods are as follows. GP-UCB, EI, and PI are classic general-purpose AFs that have been shown to achieve good regret performance for black-box functions drawn from GP. For GP-UCB, we tune its exploration parameter $\delta$ by a grid search between $10^{-1}$ and $10^{-6}$. Among the family of entropy search methods, MES is a strong and computationally efficient benchmark method that achieves superior performance for several global optimization benchmark functions and GP functions [12]. For MES, we use Gumbel sampling and take one sample for $y_*$, as suggested by the original paper [12]. TAF uses metadata to construct a transfer acquisition function by forming a mixture of experts based on an ensemble of GP models. For TAF, we consider a mixture of five experts and use the same meta-data as FSAF. Speartmint is a popular and highly optimized BO framework proposed by [1]. For Spearmint, we use the source code[2] and the default setting provided by [1]. HEBO is one recent general-purpose BO approach that addresses the heteroscedasticity and non-stationarity of black-box functions, and we use the source code[3] and the default configuration suggested by [39]. MetaBO is a neural AF trained via policy-based RL and recently achieves superior performance in BO. For fair and reproducible evaluations, we use the pre-trained model of MetaBO provided by [27]. As the original MetaBO does not address the use of meta-data, for a more comprehensive comparison, we further consider a few-shot variant of MetaBO (termed MetaBO-T), which is obtained by performing 100 more training iterations on the pre-trained MetaBO model using the meta-data for few-shot fine-tuning.

**Configuration of FSAF.** For training, we construct a collection of training tasks, each of which is a class of GP functions with either an RBF, Matern-3/2, or a spectral mixture kernel with different parameters (e.g., lengthscale and periods). For the reward design of FSAF, we use $g(z) = -\log z$ to encourage high-accuracy solutions. We choose $N = 5$, $K = 1$, and $S = 1$ given the limitation of GPU memory. For testing, we use the model with the best average total return during training as our initial model, which is later fine-tuned via few-shot fast adaptation for each task. For a fair comparison, we ensure that FSAF and MetaBO-T use the same amount of meta-data in each experiment. The source code for our experiments has been made publicly available[4].

---

[2]The source code is obtained from `https://github.com/JasperSnoek/spearmint`.

[3]HEBO is the winner of NeurIPS 2020 Black-Box Optimisation Challenge. The source code is obtained from `https://github.com/huawei-noah/HEBO`.

[4]`https://github.com/pinghsieh/FSAF`.

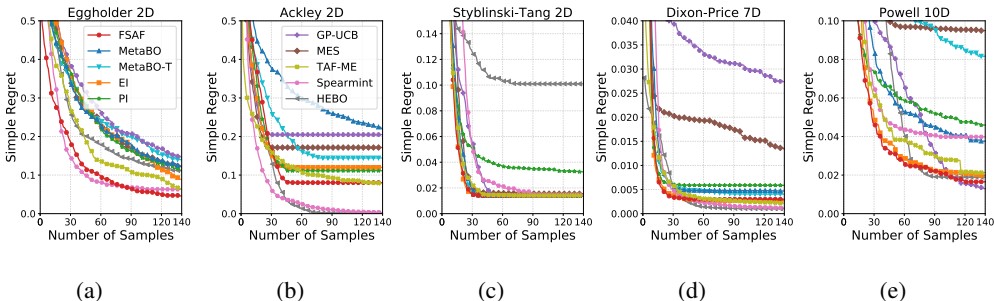

Figure 2: Mean simple regrets of FSAF and the benchmark methods under optimization test functions.

**Does FSAF adapt effectively to a wide variety of black-box functions?** To answer this, we first evaluate FSAF and the benchmark methods on several types of standard optimization benchmark functions, including: (i) *Ackley* and *Eggholder*, which are functions with a large number of local optima but with different symmetry structures; (ii) *Dixon-Price*, a valley-shaped function; (iii) *Styblinski-Tang*, a smooth function with a couple of local optima; (iv) *Powell*, a 10-dimensional function (the highest input dimension among the five functions). As the $y$ values of these functions can be one or more orders of magnitude different from each other, for ease of comparison, we scale all the values of the functions to $[-2, 2]$. Such scaling still preserves the salient structure and variations of each test function. To construct the training and testing sets, we apply random translations and re-scalings of up to +/- 10% to $x$ and $y$ values, respectively. In this case, we consider 5-shot adaptation for FSAF and use the same amount of meta-data for MetaBO-T. From Figure 2, we observe that FSAF is constantly the best or among the best of all the methods under all the test functions. We observe that MetaBO performs poorly under functions with many local optima but performs better under smooth functions (e.g., Styblinski-Tang and Powell). This might be due to the fact that MetaBO was trained with smooth GP functions and lacks the ability to adapt to functions with more local variations. We also find that MetaBO-T benefits from meta-data and improves upon MetaBO in some functions. While this manifests the potential benefits of using meta-data for MetaBO, this also suggests that brute-force fine-tuning is not effective and a more careful design like FSAF is needed. Spearmint performs well under the lower-dimensional functions (e.g., Eggholder and Ackley) but has high regrets under the higher-dimensional Powell function. This appears consistent with the results in [40]. HEBO is strong under Powell and Ackley but suffers under Eggholder and Styblinski-Tang. Moreover, among the benchmark methods, the best-performing AF indeed varies under different types of functions. This corroborates the commonly-seen phenomenon and our motivation.

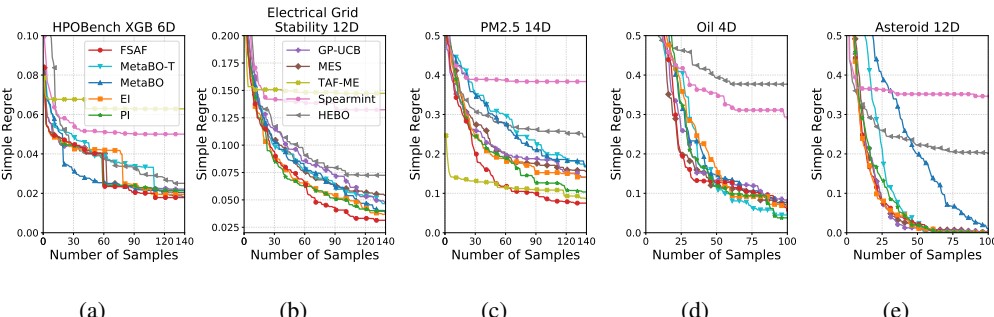

Figure 3: Mean simple regrets of FSAF and other benchmark methods under real-world test functions.

We proceed to evaluate FSAF on test functions obtained from five open-source real-world tasks in different application domains. Based on the smoothness characteristics[5], the datasets can be categorized as: (i) Smooth in all dimensions: Asteroid size prediction ($d = 12$); (ii) Smooth in all but one dimension: hyperparameter optimization for XGBoost ($d = 6$) and air quality prediction in PM 2.5 ($d = 14$); (iii) Smooth in about half of the dimensions: maximization of electric grid stability

---

[5]To better understand the characteristics of each real-world dataset, we extract the smoothness information through marginal likelihood maximization on a surrogate GP model with an RBF kernel.

($d = 12$); (iv) Non-smooth in all dimensions: location selection for oil wells ($d = 4$). The detailed description of the datasets is in Appendix C. In this setting, we consider 1-shot adaptation for FSAF, a rather sample-efficient scenario of few-shot learning. From Figure 3, we observe that FSAF remains the best or among the best for all the five real-world test functions, despite the salient structural differences of the datasets. Based on the above discussion, we confirm that FSAF indeed achieves favorable adaptability. More experimental details are provided in the supplementary material.

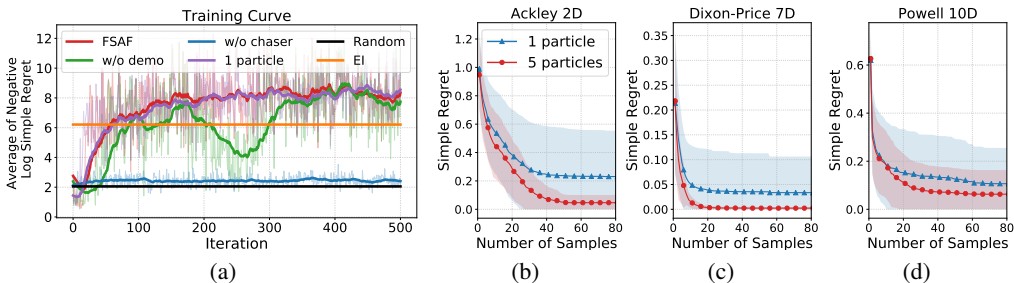

Figure 4: Ablation study for FSAF: (a) Training curves of FSAF and its ablations (EI and Random as baselines); (b)-(d) Testing performance of FSAF with 1 and 5 particles of Q-networks.

**Does FSAF mitigate the overfitting issue?** To answer this, we show the training curves of FSAF and its ablations, including: (i) FSAF without using the demo replay buffer (termed "w/o demo"); (ii) FSAF with the chaser meta-loss replaced by TD loss (termed "w/o chaser"); (iii) FSAF with only 1 particle (termed "1 particle"), which is equivalent to DQN+MAML with an additional demo replay buffer. The training metric plotted is the negative logarithm of simple regret at $t = 30$ (averaged over episodes in each iteration), which is consistent with the rewards of FSAF and can better demonstrate the differences in training. The results of EI and random sampling are given as references.

- **The Bayesian variant of DQN does mitigate overfitting.** This is confirmed by the fact that the training curves of FSAF with 1 and 5 particles are quite close, while in Figures 4(b)-4(d) the testing performance of FSAF with 5 particles appears much better than that of 1 particle.
- **The demo-based prior helps improve the training stability.** This is verified by that without the demo replay buffer, the training progress becomes apparently slower initially and is subject to more variations throughout the training.
- **The chaser meta-loss appears effective in FSAF.** Interestingly, we find that chaser meta-loss results in stable and effective training, while the TD meta-loss can barely make any progress.

**Does FSAF benefit from the few-shot gradient updates?** To better understand the effect of few-shot gradient updates, Figure 5 shows the simple regrets of FSAF under different number of gradient updates (i.e., $K$ defined in Section 3.3) for the optimization benchmark functions. We find that the adaptation effect does increase with $K$ for small $K$'s for most of the cases. This appears consistent with the general observations of MAML-like algorithms [18].

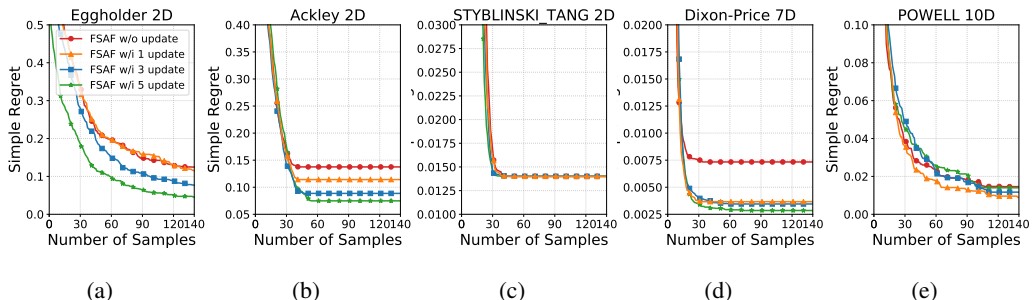

Figure 5: Simple regrets of FSAF vs number of gradient steps for optimization benchmark functions.

**Remark 8 (Application scopes of the benchmark methods)** Recall that FSAF is positioned as a meta-learning BO algorithm that can leverage a small amount of metadata for few-shot fast adaptation. In contrast, MetaBO is not designed for few-shot fast adaptation but for transfer learning in BO. Despite the difference, we choose MetaBO and its fine-tuned version MetaBO-T as important

benchmark methods since there are not many meta-learning BO algorithms and MetaBO appears to be a strong benchmark method in the class of meta-learning for BO [27]. On the other hand, while the two important benchmark methods Spearmint and HEBO do not address fast adaptation with metadata, they share the application scope of optimizing general black-box functions with FSAF.

**Remark 9 (FSAF and [15])** As mentioned in Section 1, [15] proposes to leverage meta-data to fine-tune the initialization of the GP kernel parameters for the off-the-shelf AFs (called FSBO in [15]). By contrast, FSAF uses meta-data for fast adaptation of an AF, which is a direction orthogonal to FSBO. Moreover, it is possible for our FSAF and FSBO to complement each other. We provide experimental results to verify this argument in Appendix D.

## 5 Related Work

**Few-shot learning.** Few-shot learning has recently attracted much attention since the data scarcity has become a bottleneck to many real-world machine learning tasks [41]. Using prior knowledge, few-shot learning can rapidly generalize a pre-trained model to new tasks given only a few examples. One of the most representative frameworks for few-shot learning is MAML [18], where an initialization of parameters is pre-trained to be close to the tasks drawn from the task distribution, and thus a few gradient steps are sufficient to adapt it to a specific task. There are several follow-up studies for MAML. For instance, in [42], vanilla MAML is shown to be sensitive to the network hyperparameters, often leading to training instability. To overcome that, learning the majority of hyperparameters end to end is proposed. In [20], a Bayesian version of MAML is proposed to learn complex uncertainty structure beyond a simple Gaussian approximation. In [43], MAML is revisited under multimodel task distributions. To better adapt to different modes, a modulation network is proposed to identify the mode of the task distribution and then customize the meta-learned prior for the identified mode.

**Bayesian optimization via non-myopic methods.** To go beyond myopic AFs, recently there has been some interest in designing non-myopic strategies for BO from the perspective of dynamic programming (DP). For example, [44, 45] characterized the optimal non-myopic strategies for BO. To tackle the intractable DP problem, [46, 47] proposed to approximately solve the DP problem for BO by different simulation techniques. [48] focused on the two-step lookahead AFs for BO and proposed an efficient Monte-Carlo method to find such AFs. Instead of fixing the horizon in advance, [49] proposed a principled way to select the rolling horizon for approximate dynamic programming in BO. [50] proposed to achieve non-myopia by maximizing a lower bound on the multi-step expected utility. [51] proposed a tree-based AF to enable approximate multi-step lookahead in a one-shot manner by leveraging the reparameterization trick. The proposed FSAF is also non-myopic by nature as it implicitly takes multi-step effect into account by using Q-networks as AFs. Different from the above multi-step solutions, FSAF is meant to serve as an AF that can adapt to a wide variety of black-box functions given only a small amount of meta-data.

**Meta Bayesian optimization.** Meta-learning for BO has been discussed in many prior studies. Some of them [52, 53] use a single GP model as the prior of all the tasks. However, in real-world applications, different tasks may have different task-specific features that fail to be captured by a single model. To mitigate the issue, several studies propose to learn an ensemble of GPs as the prior. For instance, Wistuba et al. [16] proposed to use a transferable AF that uses several GPs to evaluate the target dataset then weights expected improvement by the result of the evaluation for fitting in the current task. Feurer et al. [17] proposed to use an ensemble of GP models obtained from each subset of meta-data and use all of them for inference in the new task. But these works still need a sufficiently large amount of data for knowledge transfer to new tasks. [40] proposed to train a general-purpose black-box function optimizer by using recurrent neural networks. MetaBO [27] used policy-based RL to train a neural AF that learns structural properties of a set of source tasks to enable knowledge transfer to related new tasks. FSBO [15] provides a few-shot deep kernel network for a GP surrogate that can quickly adapt its kernel parameters to unseen tasks. However, the benefits of using meta-data for more efficient exploration via few-shot fast adaptation of AFs were not explored by [15, 27, 40].

## 6 Concluding Remarks

This paper tackles the critical challenge of how to effectively adapt an AF for BO to a wide variety of black-box functions through the lens of few-shot acquisition function (FSAF) learning. One potential limitation of FSAF is the need of a small amount of meta-data. Without any few-shot adaptation, the performance may not always be satisfactory, as shown in Figure 5. Despite this, through extensive experiments, we show that FSAF is indeed a promising general-purpose approach for BO.

## Acknowledgment

The work of Bing-Jing Hsieh and Ping-Chun Hsieh is supported in part by the Ministry of Science and Technology (MOST) of Taiwan under Contract Numbers MOST 109-2636-E-009-012 and MOST 110-2628-E-A49-014.

## Funding Transparency Statement

Funding in direct support of this work: MOST grants under Contract Numbers MOST 109-2636-E-009-012 and MOST 110-2628-E-A49-014, awarded to Ping-Chun Hsieh in National Yang Ming Chiao Tung University. There is no additional revenue related to this work.

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
