# Appendix

## A Pseudo Code and Architecture of the FSAF Training Algorithm

For completeness, we provide the pseudo code of the training algorithm of FSAF in the following Algorithm 1. Figure 6 further illustrates the demo policy and the replay buffers used in FSAF.

---

**Algorithm 1** FSAF Training Algorithm

---

1: **Initialize:** $N$ instances of Q-networks with parameters $\Theta = \{\theta^{(n)}\}_{n=1}^{N}$, demo policy $\pi_{\mathrm{D}}$, a collection of candidate tasks $\mathcal{T}$, and the parameters $K, S$ for computing the meta-loss
2: **for** each iteration $i = 0, 1, \cdots$ **do**
3:      Sample a batch of tasks $\mathcal{T}_i$ from $\mathcal{T}$
4:      **for** each task $\tau \in \mathcal{T}_i$ **do**
5:          Generate black-box functions of task $\tau$ from GP and set $\Theta_{\tau,0} = \Theta$
6:          **for** $k = 0, \cdots, K-1$ **do**
7:              Collect trajectories $\{(s_0, a_1, r_1, \cdots)\}$ using $\pi_{\Theta_{\tau,k}}$ and store transitions in $\mathcal{R}_Q$
8:              Collect trajectories $\{(s_0, a_1, r_1, \cdots)\}$ using $\pi_{\mathrm{D}}$ and store transitions in $\mathcal{R}_{\mathrm{D}}$
9:              Sample a mini-batch of transitions $\mathcal{D}_{\tau,k}^{\mathrm{tr}}$ from the replay buffers $\mathcal{R}_Q, \mathcal{R}_{\mathrm{D}}$
10:              $\Theta_{\tau,k+1} = \mathcal{M}_{\mathrm{SVGD}}(\Theta_{\tau,k}, \mathcal{D}_{\tau,k}^{\mathrm{tr}})$
11:          **end for**
12:          Set $\Theta_{\tau,0}^{*} = \Theta_{\tau,K}$
13:          **for** $s = 0, \cdots, S-1$ **do**
14:              Sample a mini-batch of transitions $\mathcal{D}_{\tau,s}^{\mathrm{val}}$ from the replay buffers $\mathcal{R}_Q, \mathcal{R}_{\mathrm{D}}$
15:              $\Theta_{\tau,s+1}^{*} = \mathcal{M}_{\mathrm{SVGD}}(\Theta_{\tau,s}^{*}, \mathcal{D}_{\tau,s}^{\mathrm{val}})$
16:          **end for**
17:      **end for**
18:      $\Theta \leftarrow \Theta - \beta \nabla_{\Theta}\left(\sum_{\tau \in \mathcal{T}_i} \mathcal{L}_{\mathrm{meta}}(\Theta; \tau)\right)$
19: **end for**

---

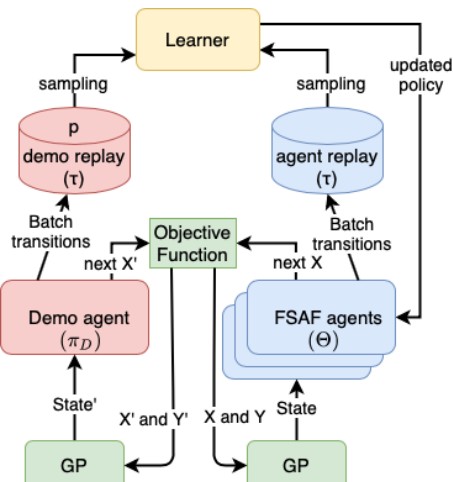

Figure 6: Architecture of the proposed FSAF.

## B Detailed Training Configuration of FSAF

**Training tasks of FSAF.** To construct a diverse collection of tasks for the training of FSAF, we leverage GP functions with RBF, Matèrn-3/2, and spectral mixture kernels to capture smooth functions, functions with abrupt local variations, and functions of periodic nature, respectively. For both the RBF and the Matèrn-3/2 kernels, we consider three possible ranges of lengthscales, including $[0.07, 0.13], [0.17, 0.23], [0.27, 0.33]$. For the spectral mixture kernels, we consider mixtures of two Gaussian components of periods $0.3$ and $0.6$ and three possible ranges of the lengthscales, including

$[0.27, 0.33], [0.47, 0.53], [0.57, 0.63]$. As a result, there are nine candidate tasks in the task collection $\mathcal{T}$. In each training iteration $i$, three out of the nine tasks are selected uniformly at random from the above task collection (and hence $|\mathcal{T}_i| = 3$ in Algorithm 1). The input domain of these GP functions is configured to be $[0, 1]^3$. To facilitate the training procedure, we discretize the continuous input domain by using the Sobol sequence to generate a grid on which the GP functions and the AFs are evaluated, as typically done in BO.

**Demo policy for the prior of Bayesian DQN.** Recall from Section 3.2 that we leverage a Bayesian variant of DQN and construct an informative prior using a demo policy induced by an off-the-shelf AF. For the demo policy, we use EI, which is computationally efficient and achieves moderate regrets in most of the cases. To control how much the demo policy involves in the training, we use a hyperparameter termed *demonstration ratio* $\kappa$, which is implemented by randomly choosing whether to use a mini-batch of transitions from the demo replay buffer with probability $\kappa$ at each SVGD step. In our experiments, we find that a small demonstration ratio of $\frac{1}{128}$ is sufficiently effective.

**Hyperparameters of Bayesian DQN.** Recall the hyperparameters $N$, $K$, and $S$ defined in Algorithm 1. We take $N = 5$, $K = 5$, and $S = 1$ given the memory limitation of GPUs. Despite the small values of $N$, $K$, and $S$, these choices already provide superior performance.

**Network architecture of each DQN particle.** For all the DQN particles used in the experiments, we adopt the standard dueling network architecture [54], where one value network and an advantage network are maintained to produce the estimated Q-values. For a fair comparison between FSAF and MetaBO, both the value network and the advantage network of our FSAF are configured to have 4 fully-connected hidden layers with ReLU activation functions and 200 hidden units per layer. As described in Section 3.1, the input of an advantage network consists of a four-tuple, namely the posterior mean $\mu_t(x)$, the posterior standard deviation $\sigma_t(x)$, the best observation so far $y_t^*$, and the ratio between the current timestamp and total sampling budget $\frac{t}{T}$. Accordingly, the input of a value network consists of $y_t^*$ and $\frac{t}{T}$.

**Computing Resources.** All the training and testing processes are run on a Linux server with (i) an Intel Xeon Gold 6136 CPU operating at a maximum clock rate of 3.7 GHz, (ii) a total of 256 GB memory, and (iii) an RTX 3090 GPU.

Table 1 summarizes the hyperparameter configuration of the training of FSAF.

Table 1: FSAF training hyperarameters.

| Description | Value |
| --- | --- |
| Batch size (i.e., $|\mathcal{D}_{\tau,k}^{\text{tr}}|$ in Algorithm 1) | 128 |
| Target update interval (in terms of iterations) | 5 |
| Lower-level learning rate (i.e., $\eta$ in (8)) | 0.01 |
| Upper-level learning rate (i.e., $\beta$ in Algorithm 1) | 0.001 |
| Agent/demo replay buffer size | 1000 |
| Discount factor $\gamma$ | 0.98 |
| Number of DQN particles | 5 |
| Total sampling budget | 100 |
| Cardinality of the Sobol grid | 200 |

## C  Experiment Details

### C.1  Experiment Details of Figure 1

Recall that Figure 1 illustrates the overfitting issue of DQN+MAML. Regarding the DQN used for Figure 1, we use the same design, network architecture, and training tasks as those described in Appendix B. Regarding the vanilla MAML for Figure 1, we use the squared TD error in (4) as the loss function for both the lower-level and upper-level updates. For both training and the fast adaptation during testing, we apply 5-shot adaptation (i.e., 5 black-box functions are used for adaptation) and set the number of few-shot gradient updates to be 5. In Figures 1(b)-1(d), we report the empirical average simple regret as well as the empirical standard deviation over 100 independent trials.

## C.2 Experiment Details of Figure 2

Recall that Figure 2 shows the simple regret performance of the AFs under various standard optimization benchmark functions. The input domain of each benchmark function is a hypercube in the form of $[-x_{\mathrm{lim}}, x_{\mathrm{lim}}]^d$ (e.g., $x_{\mathrm{lim}} = 5$ and $d = 2$ for the Ackley function). Moreover, as the function values of these benchmark functions can be one or more orders of magnitude different from each other, for ease of comparison, we scale all the values of the functions to the range $[-2, 2]$. For the posterior inference required by all the AFs, we use a GP with an RBF kernel as the surrogate model. For each benchmark function, the lengthscale parameter of the RBF kernel is estimated via marginal likelihood maximization, which is a commonly-used Bayesian model selection framework.

**Validation datasets and testing datasets.** As mentioned in Section 4, we construct the validation datasets (mainly for the few-shot adaptation of FSAF and the fine-tuning of MetaBO-T) and the testing datasets by applying random translations and re-scalings to the $x$ and the $y$ values, respectively. Specifically, the amount of translation added to each dimension of $x$ is selected from the range $[-0.1x_{\mathrm{lim}}, 0.1x_{\mathrm{lim}}]$ uniformly at random. Similarly, the re-scaling factor applied to each $y$ value is chosen uniformly at random from the range $[0.9, 1.1]$.

**Maximization of the AFs.** Recall that the input domain of each optimization benchmark function is a hypercube. To address the continuous input domains and achieve a fair comparison between FSAF and MetaBO, we leverage the hierarchical gridding method similar to that in [27] for the maximization procedure of the AFs. Specifically, we first construct a coarse Sobol grid of $N_{\mathrm{coarse}}$ points that span over the entire domain and then evaluate the AF on this grid. Next, we find the $N_{\mathrm{m}}$ maximal evaluations on this coarse grid and build a finer local Sobol grid of $N_{\mathrm{local}}$ points for each of the $N_{\mathrm{m}}$ maximal points. Then, the maximum of the AF is approximated by the maximum value among these $N_{\mathrm{m}}N_{\mathrm{local}}$ AF evaluations. To finish the testing process within a reasonable amount of time, we choose $N_{\mathrm{coarse}} = 2000$, $N_{\mathrm{m}} = 10$, and $N_{\mathrm{local}} = 1000$.

## C.3 Experiment Details of Figure 3

Recall that Figure 3 demonstrates the regret performance under the test functions obtained from a variety of real-world datasets. Below we describe these real-world datasets in more detail.

- **XGBoost Hyperparameter Optimization.** We use the HPOBench dataset[6] for the hyperparameter optimization for the XGBoost algorithm. Specifically, we use the pre-computed results of XGBoost with six tunable hyperparameters (e.g., learning rate, $L_1$ and $L_2$ regularization terms, and subsampling ratios) on 48 classification datasets, each of which is associated with 1000 randomly selected hyperparameter configurations. Hence, these 48 subsets of pre-computed results naturally provide 48 black-box test functions (with the cardinality of the input domain = 1000, for each black-box function) for BO. While these 48 black-box functions are different, they are deemed to be of common characteristics as these functions are obtained under the same XGBoost algorithm and related classification datasets. In the 1-shot setting, we use 1 out of the 48 black-box functions as the meta-data for fast adaptation of FSAF and finding the lengthscale parameter of the GP surrogate model via marginal likelihood maximization. The remaining 47 black-box functions are used only for testing. The use of meta-data is similar in other tasks.

- **Electrical Grid Stability Dataset.** This dataset corresponds to an augmented version of the *Electrical Grid Stability Simulated Dataset*[7] [55]. This dataset records total 12 features, such as the reaction times of the producer and the consumer, power balance, and price elasticity coefficient, and the objective is to maximize the grid stability. This dataset contains 60000 parameter configurations, and we divide them into 39 testing sets and 1 validation set. The validation set is used for both few-shot adaptation of FSAF as well as finding the lengthscale parameter of the GP surrogate model for posterior inference.

- **Air Quality Prediction.** We use the meteorological monitoring data of air quality collected in northern Taiwan in 2015[8]. We select 14 air quality features, such as the amount of Sulfur dioxide,

---

[6]Created by AutoML and available at `https://github.com/automl/HPOBench`.

[7]Created by Vadim Arzamasov (Karlsruher Institut für Technologie, Karlsruhe, Germany) and available at `https://www.kaggle.com/pcbreviglieri/smart-grid-stability`.

[8]Created by Environmental Protection Administration, Executive Yuan, R.O.C. (Taiwan), available at `https://airtw.epa.gov.tw/ENG/default.aspx`

Carbon monoxide, and ozone, and set the amount of $PM_{2.5}$ as the objective function. After cleaning up all the NaN entries and missing data entries, we get about 73000 parameter configurations and split them into 29 testing sets and 1 validation set. Again, the validation set is used for both few-shot adaptation of FSAF as well as finding the lengthscale parameter of the GP surrogate model for posterior inference.

- **Oil Well Dataset.** We use the *NYS Oil, Gas, Other Regulated Wells datasets*[9] to find the deepest drilled depth among the oil wells. For each data entry, we use the longitude and latitude of both the surface as well as the bottom of the oil well as the input features. This dataset contains about 41000 parameter configurations, and we divide the dataset into 29 subsets of testing data and 1 set of validation data for few-shot adaptation of FSAF and finding the lengthscale parameter of the GP surrogate model for posterior inference.

- **Asteroid Dataset.** This dataset[10] contains a variety features of the asteroids, and our goal is to find the maximum asteroid diameter based on all of its 12 numerical features. Since the range of the asteroid diameters is too large, we apply the commonly-used input warping technique in BO and use the logarithm of the diameter values as the objective. After cleaning up all the NaN entries and missing entries, we get about 136000 parameter configurations and split them into 39 testing sets and 1 validation set.

Regarding the maximization of AFs, since the input domains of the real-world test functions are all discrete, the maximum value of each AF can be found exactly and therefore the hierarchical gridding procedure is not required in this case.

## D    Additional Experimental Results

In this section, we provide additional experimental results regarding the synthetic GP test functions, the combination of FSAF and FSBO, and the percentiles associated with Figure 3 in the main text.

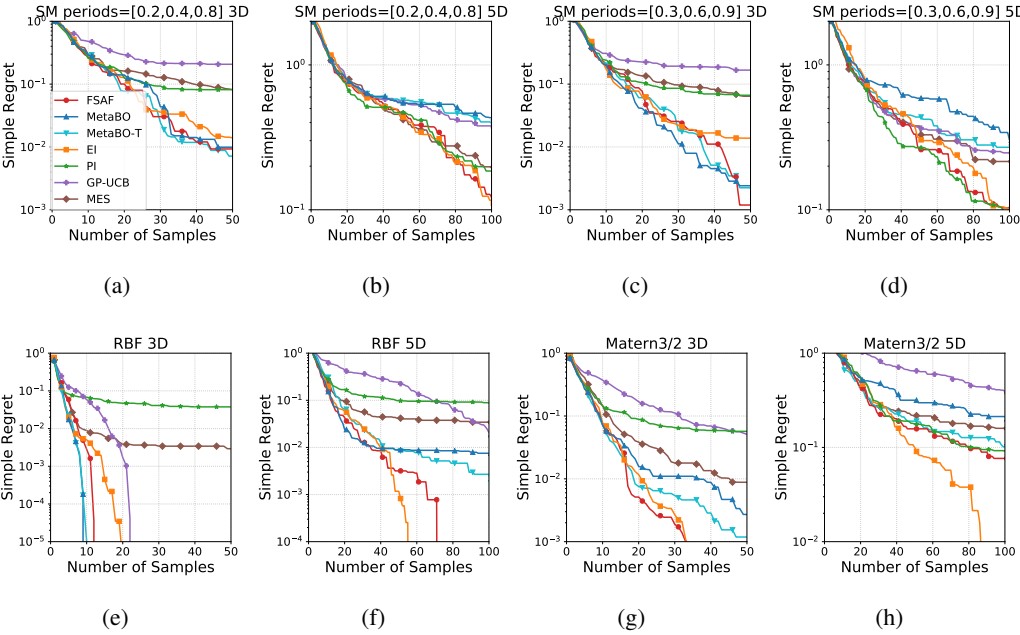

Figure 7: Median simple regrets of FSAF and other benchmark methods under synthetic GP test functions. Other percentiles are provided in Table 2 for visual clarity.

---

[9]Hosted by State of New York, available at `https://www.kaggle.com/new-york-state/nys-oil, -gas,-other-regulated-wells?select=oil-gas-other-regulated-wells-beginning-1860.csv`.

[10]Provided by the paper titled "Prediction of Asteroid Diameter with the help of Multi-layer Perceptron Regressor" in *International Conference on Computer Science, Industrial Electronics* in 2019 [56] and available at `https://www.kaggle.com/basu369victor/prediction-of-asteroid-diameter`.

### D.1 Synthetic GP Test Functions

To further validate the effectiveness of FSAF, we evaluate FSAF and the benchmark AFs on GP functions drawn from various kernels. Specifically, to generate the validation and testing datasets, we consider four different types of GP kernels for both the input domains of $[0,1]^3$ and $[0,1]^5$, including: (i) Spectral mixture kernel with three Gaussian components of periods $0.2$, $0.4$, and $0.8$ as well as a lengthscale range $[0.5, 0.55]$; (ii) Spectral mixture kernel with three Gaussian components of periods $0.3$, $0.6$, and $0.9$ as well as a lengthscale range $[0.5, 0.55]$ (iii) RBF kernel with a lengthscale range $[0.5, 0.55]$; (iv) Matèrn kernel with a lengthscale range $[0.5, 0.55]$. Notably, all the above kernels have never been seen by FSAF during training. To address the continuous input domains, we use the same hierarchical gridding method as described in Appendix C.2 to maximize the AFs. Similar to the case of Figure 2, we consider 5-shot adaptation for FSAF and use the same amount of meta-data for MetaBO-T.

Figure 7 shows the simple regrets of all the AFs under eight different types of GP functions. Again, we can see that FSAF is constantly among the best of all the methods under most of the test functions. We also observe that MetaBO and MetaBO-T perform quite well in the 3-dimensional tasks. We conjecture that this is due to the fact that the pre-trained MetaBO model was originally trained on 3-dimensional GP functions, as described in [27]. However, MetaBO and MetaBO-T do not adapt well to the 5-dimensional GP functions, as shown in Figures 7(b), 7(d), 7(f), and 7(h). By contrast, despite that FSAF is also trained on 3-dimensional GP functions, FSAF adapts more effectively to GP functions of a higher input dimension, as shown in 7(b), 7(d), 7(f). On the other hand, Figure 7(h) shows that FSAF does not catch up well with EI after 40 samples under the 5-dimensional Matèrn-2/3 functions. We conjecture that this is due to the fact that 5-dimensional Matèrn-2/3 functions have even more local variations that could not be captured well by only a small amount of meta-data. Moreover, among the benchmark methods, the best-performing AF still varies under different types of functions. This again corroborates the commonly-seen phenomenon and our motivation.

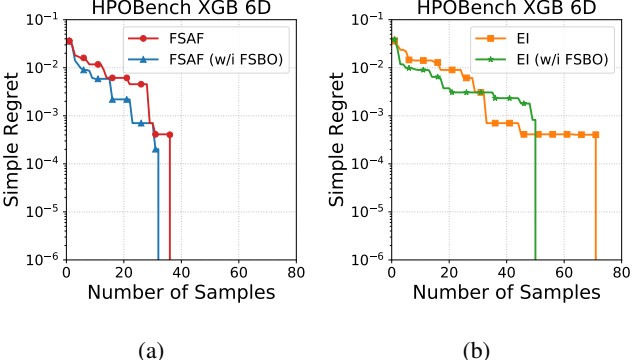

(a)            (b)

Figure 8: Median simple regrets of FSAF and EI, with and without the integration with FSBO. Other percentiles are in Table 3.

### D.2 Combining FSAF with FSBO

Recall from Remark 9 that [15] proposes FSBO, which leverages meta-data to fine-tune the initialization of the GP kernel parameters for the off-the-shelf AFs. As FSBO can provide better GP kernel parameters for posterior inference than the conventional approaches (e.g., marginal likelihood maximization), it appears feasible to let FSAF and FSBO complement each other and even combine other off-the-shelf AFs with FSBO for better regret performance. In this section, we provide experimental results to validate the above argument. Since [15] did not release their source code, we need to re-implement FSBO by ourselves[11]. Specifically, for the meta-learning part of FSBO, we leverage Reptile [57] with a spectral mixture kernel as a lightweight variant of [58]. Moreover,

---

[11]We contacted the authors of [15] via email and got the reply that they still needed a bit more time before making the code publicly available. Moreover, we also confirmed the important design choices with the authors to better reproduce FSBO.

Table 2: The percentiles of simple regrets over 100 independent trials for all the AFs under synthetic GP test functions. For 3-dimensional kernel functions, we report the percentiles measured at steps 25 and 50. For 5-dimensional kernel functions, we report the percentiles measured at steps 50 and 100. All the values displayed here are scaled by 100 for more compact notations. The best of each percentile under each type of functions is highlighted in **bold and underline**.

| AF | P | SM periods= [0.2,0.4,0.8] 3D | SM periods= [0.3,0.6,0.9] 3D | RBF 3D | Matèrn-3/2 3D |
|---|---|---|---|---|---|
| FSAF | 25% | 0.03, **0.00** | 0.01, **0.00** | **0.00**, **0.00** | **0.00**, **0.00** |
| | 50% | 7.80, 0.91 | 3.28, **0.12** | **0.00**, **0.00** | **0.25**, 0.03 |
| | 75% | 41.62, 38.81 | **24.78**, 16.74 | **0.00**, **0.00** | **3.87**, 0.54 |
| | 90% | **70.97**, 67.15 | 64.65, 62.63 | **0.55**, 0.52 | 16.75, 9.67 |
| MetaBO | 25% | **0.00**, 0.00 | **0.00**, 0.00 | **0.00**, 0.00 | 0.02, **0.00** |
| | 50% | 10.88, 0.99 | **2.19**, 0.24 | **0.00**, **0.00** | 1.10, 0.27 |
| | 75% | 46.07, 29.76 | 28.96, 17.16 | 0.19, 0.13 | 11.51, 5.77 |
| | 90% | 81.47, 75.46 | 72.62, 61.25 | 0.92, 0.84 | 25.12, 22.52 |
| MetaBO-T | 25% | 0.09, **0.00** | 0.15, **0.00** | **0.00**, 0.00 | **0.00**, 0.00 |
| | 50% | 6.47, **0.72** | 3.97, 0.22 | **0.00**, **0.00** | 0.65, 0.12 |
| | 75% | 46.25, 16.45 | 41.52, 18.71 | 0.25, 0.02 | 7.04, 4.93 |
| | 90% | 82.98, 51.48 | 78.32, 59.29 | 1.41, 0.84 | 24.48, 23.85 |
| EI | 25% | 1.02, 0.50 | 0.93, 0.34 | **0.00**, 0.00 | 0.00, **0.00** |
| | 50% | **4.57**, 1.22 | 2.60, 1.20 | **0.00**, **0.00** | 0.36, **0.00** |
| | 75% | 49.63, **6.27** | 29.60, **4.19** | 0.33, 0.02 | 5.31, **0.43** |
| | 90% | 88.47, 48.41 | 83.11, 38.15 | 1.22, 0.99 | 14.93, **1.38** |
| PI | 25% | 5.93, 3.96 | 4.55, 3.54 | 1.06, 1.01 | 3.94, 3.21 |
| | 50% | 10.34, 8.16 | 8.76, 6.65 | 4.59, 3.76 | 7.26, 5.70 |
| | 75% | 40.42, 14.74 | 28.49, 12.45 | 8.04, 6.71 | 12.61, 9.13 |
| | 90% | 79.30, 42.57 | 74.08, 33.11 | 11.45, 8.93 | 19.77, 13.93 |
| GP-UCB | 25% | 13.04, 11.71 | 11.18, 10.80 | **0.00**, **0.00** | 6.06, 1.79 |
| | 50% | 23.83, 20.70 | 20.46, 16.71 | **0.00**, **0.00** | 13.20, 5.11 |
| | 75% | 47.45, 33.06 | 35.04, 27.53 | 0.21, **0.00** | 26.47, 9.42 |
| | 90% | 73.17, 47.62 | **59.32**, 34.97 | 1.11, **0.44** | 41.64, 16.09 |
| MES | 25% | 5.70, 3.70 | 5.72, 2.66 | 0.03, 0.02 | 0.69, 0.21 |
| | 50% | 14.97, 8.18 | 10.69, 6.42 | 0.37, 0.29 | 3.05, 0.88 |
| | 75% | **37.36**, 16.17 | 28.17, 10.70 | 1.81, 1.18 | 7.59, 2.88 |
| | 90% | 75.31, **36.15** | 67.86, **17.39** | 5.03, 3.75 | **11.78**, 6.27 |
| AF | P | SM periods= [0.2,0.4,0.8] 5D | SM periods= [0.3,0.6,0.9] 5D | RBF 5D | Matèrn-3/2 5D |
| FSAF | 25% | 8.10, 0.88 | 5.51, 1.75 | **0.00**, **0.00** | 2.67, **0.00** |
| | 50% | 46.04, 12.03 | **25.44**, 10.08 | 0.31, **0.00** | 15.40, 7.47 |
| | 75% | **67.89**, **46.69** | 74.85, 41.58 | **2.67**, **1.32** | 45.85, 23.46 |
| | 90% | **104.46**, **71.09** | **98.25**, 78.89 | **5.52**, **4.00** | 72.67, 52.40 |
| MetaBO | 25% | 22.55, 13.76 | 14.32, 6.84 | **0.00**, **0.00** | 9.18, 3.44 |
| | 50% | 54.93, 43.24 | 58.90, 30.21 | 0.89, 0.75 | 31.69, 21.27 |
| | 75% | 80.30, 69.47 | 101.54, 78.60 | 7.02, 4.92 | 69.30, 48.99 |
| | 90% | 128.87, 96.96 | 156.15, 124.84 | 30.42, 17.31 | 86.73, 80.38 |
| MetaBO-T | 25% | 23.23, 15.84 | 12.31, 5.69 | **0.00**, **0.00** | 4.82, 0.61 |
| | 50% | 57.05, 39.99 | 43.67, 27.00 | 0.85, 0.27 | 19.01, 10.17 |
| | 75% | 88.13, 69.88 | 75.39, 65.34 | 4.58, 3.12 | 55.67, 28.54 |
| | 90% | 123.23, 91.40 | 107.24, 92.93 | 23.03, 8.77 | 77.27, 55.63 |
| EI | 25% | **6.35**, 0.00 | **3.12**, 0.00 | **0.00**, 0.00 | **0.51**, 0.00 |
| | 50% | **38.09**, **10.12** | 27.56, **8.42** | 0.10, **0.00** | **8.68**, **0.00** |
| | 75% | 79.77, 51.62 | 86.52, 57.03 | 3.39, 1.59 | **28.38**, **7.36** |
| | 90% | 143.68, 92.64 | 123.61, 102.67 | 8.09, 5.56 | 68.32, 30.38 |
| PI | 25% | 12.74, 4.36 | 6.76, 2.45 | 5.81, 4.80 | 8.54, 4.42 |
| | 50% | 46.90, 18.52 | 26.81, 9.55 | 9.76, 8.80 | 17.61, 9.13 |
| | 75% | 76.23, 53.18 | 74.40, **40.77** | 14.47, 12.75 | 30.70, 17.44 |
| | 90% | 110.59, 89.79 | 107.18, **70.79** | 20.94, 16.77 | **53.25**, **27.73** |
| GP-UCB | 25% | 24.86, 17.37 | 21.25, 15.13 | 12.79, **0.00** | 41.12, 19.84 |
| | 50% | 54.77, 37.94 | 37.17, 24.71 | 24.31, 2.06 | 64.86, 39.35 |
| | 75% | 88.69, 64.84 | **71.08**, 46.74 | 37.09, 7.72 | 84.72, 56.06 |
| | 90% | 127.81, 92.97 | 105.94, 71.43 | 46.44, 13.06 | 111.00, 68.07 |
| MES | 25% | 19.37, 4.40 | 11.08, 5.16 | 0.84, 0.43 | 9.31, 4.46 |
| | 50% | 42.62, 19.73 | 37.91, 21.55 | 4.17, 3.43 | 21.72, 15.95 |
| | 75% | 74.29, 46.93 | 73.01, 50.09 | 11.76, 9.91 | 40.49, 28.11 |
| | 90% | 109.83, 72.50 | 98.32, 85.27 | 18.17, 16.03 | 67.14, 59.04 |

we use cosine annealing outer-loop learning rate from $10^{-3}$ to $10^{-5}$ and set the inner-loop learning rate to be $10^{-2}$. The deep kernel is represented by a neural network with two hidden layers (with 128 hidden units per layer), and the degree of few-shot deep kernel learning is configured to be 4. The spectral mixture kernel is configured to have 10 components. We test (i) the combination of FSAF and FSBO as well as (ii) the combination of EI and FSBO on the XGBoost hyperparameter optimization task described in Appendix C.3. As FSBO requires some data for obtaining an initial model, we use 5 out of the 48 subsets of the HPOBench dataset to obtain an initial FSBO model and evaluate the regret performance on the other 43 subsets. From Figure 8 and Table 3, we see that FSBO can slightly improve the regret performance of the two AFs.

Table 3: The percentiles of simple regrets under the HPOBench dataset at steps 30 and 60. All the values displayed here are scaled by 100 for more compact notations. The best of each percentile under each type of functions is highlighted in **bold and underline**.

| Dataset | Percentile | FSAF | FSAF (w/i FSBO) |
|---|---|---|---|
| HPOBench XGB | 25% | **0.00**, **0.00** | **0.00**, **0.00** |
| | 50% | **0.07**, **0.00** | **0.07**, **0.00** |
| | 75% | 1.79, **0.90** | **0.99**, 0.99 |
| | 90% | 4.51, **3.31** | **3.37**, 3.37 |

| Dataset | Percentile | EI | EI (w/i FSBO) |
|---|---|---|---|
| HPOBench XGB | 25% | **0.00**, **0.00** | **0.00**, **0.00** |
| | 50% | **0.31**, 0.04 | **0.31**, **0.00** |
| | 75% | 2.50, 1.99 | **1.57**, **1.13** |
| | 90% | 6.36, 6.27 | **5.18**, **4.62** |

## D.3 Additional Percentiles for Figure 3

In this section, we provide more detailed percentiles associated with the experiments of Figure 3 in Table 4. We can see that FSAF is still among the best in terms of either lower or higher percentiles under most of the real-world test functions.

Table 4: The percentiles of simple regrets for all the AFs under different real-world test functions. For PM2.5, and Electrical Grid Stability, we report the percentiles measured at steps 60 and 120. For HPOBench XGB, Oil, and Asteroid, we report the percentiles measured at steps 30 and 60. All the values displayed here are scaled by 100 for more compact notations. The best of each percentile under each type of functions is highlighted in **bold and underline**.

| AF | P | Electrical Grid Stability | PM2.5 | HPOBench XGB | Oil | Asteroid |
|---|---|---|---|---|---|---|
| FSAF | 25% | **0.00**, **0.00** | **0.00**, **0.00** | **0.00**, **0.00** | **0.00**, **0.00** | **0.00**, **0.00** |
|  | 50% | **4.48**, **0.00** | **6.00**, **1.00** | **0.14**, **0.00** | **16.59**, **0.00** | 2.85, **0.00** |
|  | 75% | 11.92, **5.76** | 19.00, **11.00** | 3.16, 2.03 | 40.82, 16.59 | 11.16, **0.00** |
|  | 90% | 15.03, **11.78** | 28.80, 22.20 | 8.42, 6.29 | 50.89, **22.89** | 18.90, **0.00** |
| MetaBO | 25% | 2.42, **0.00** | 20.00, 9.00 | **0.00**, **0.00** | **0.00**, **0.00** | 16.66, **0.00** |
|  | 50% | 7.43, 2.70 | 26.00, 17.00 | 1.38, 0.28 | 20.28, 1.96 | 40.58, 10.54 |
|  | 75% | 11.15, 6.86 | 39.00, 26.00 | **3.15**, **1.51** | 43.34, 25.69 | 50.32, 19.76 |
|  | 90% | 17.59, 13.61 | 44.40, 35.00 | **6.62**, 5.79 | 60.52, 37.25 | 58.37, 35.71 |
| MetaBO-T | 25% | 2.81, **0.00** | 23.00, 12.00 | **0.00**, **0.00** | 15.96, **0.00** | **0.00**, **0.00** |
|  | 50% | 6.86, 3.74 | 30.00, 16.00 | 1.38, 0.61 | 31.71, **0.00** | 2.58, **0.00** |
|  | 75% | 12.17, 9.72 | 39.00, 28.00 | 3.46, 2.47 | 49.05, 15.75 | 22.93, **0.00** |
|  | 90% | 14.76, 13.89 | 43.60, 36.00 | 8.27, 6.62 | 63.61, 31.67 | 32.49, **0.00** |
| EI | 25% | 0.30, **0.00** | 12.00, 5.00 | **0.00**, **0.00** | 15.10, **0.00** | **0.00**, **0.00** |
|  | 50% | 5.72, **0.00** | 16.00, 15.00 | 0.60, 0.04 | 25.60, **0.00** | 0.00, **0.00** |
|  | 75% | **9.84**, 7.12 | 29.00, 21.00 | 3.16, 2.70 | 43.43, **15.63** | 8.73, **0.00** |
|  | 90% | **14.66**, 12.72 | 36.00, 33.60 | 8.03, 8.03 | 63.95, 25.83 | 21.64, **0.00** |
| PI | 25% | **0.00**, **0.00** | 6.00, **0.00** | **0.00**, **0.00** | 0.27, **0.00** | **0.00**, **0.00** |
|  | 50% | 5.72, 1.30 | 17.00, 8.00 | 0.60, 0.03 | 18.76, 1.66 | 2.85, **0.00** |
|  | 75% | **9.84**, 7.30 | 33.00, 15.00 | 3.16, 2.14 | 29.18, 15.75 | 17.60, **0.00** |
|  | 90% | 15.31, 12.48 | 37.40, 32.20 | 8.03, **4.92** | 53.95, 27.49 | 23.13, **0.00** |
| GP-UCB | 25% | 1.49, **0.00** | 11.00, 8.00 | **0.00**, **0.00** | 14.11, **0.00** | **0.00**, **0.00** |
|  | 50% | 8.05, 2.87 | 18.00, 15.00 | 0.60, 0.04 | 21.54, 13.27 | 0.00, **0.00** |
|  | 75% | 14.55, 9.84 | 28.00, 20.00 | 3.16, 2.53 | 30.98, 21.83 | **4.84**, **0.00** |
|  | 90% | 17.93, 12.83 | 38.20, 30.40 | 8.03, **4.92** | 49.56, 30.39 | 20.66, 1.97 |
| MES | 25% | 1.49, **0.00** | 11.00, 8.00 | **0.00**, **0.00** | 1.95, **0.00** | **0.00**, **0.00** |
|  | 50% | 6.81, 2.89 | 15.00, 14.00 | 0.61, 0.04 | 20.28, 1.95 | 1.36, **0.00** |
|  | 75% | 13.61, 9.17 | 32.00, 16.00 | 3.16, 2.73 | 28.83, 20.93 | 7.62, **0.00** |
|  | 90% | 16.83, 13.22 | 37.00, 37.00 | 8.03, 6.29 | **33.85**, 30.42 | **14.82**, 3.12 |
| TAF-ME | 25% | 9.18, 9.18 | 4.00, 2.00 | 0.06, 0.02 | 0.27, 0.27 | **0.00**, **0.00** |
|  | 50% | 15.86, 15.86 | 9.00, 6.00 | 2.50, 2.48 | **16.59**, 14.63 | 3.75, 2.58 |
|  | 75% | 19.78, 19.56 | **18.00**, 15.00 | 8.09, 6.98 | **25.60**, 20.28 | 13.17, 12.31 |
|  | 90% | 23.84, 23.84 | **22.60**, **21.00** | 15.77, 14.00 | 45.90, 26.82 | 21.58, 20.25 |
| Spearmint | 25% | 8.48, 8.48 | 29.50, 29.50 | **0.00**, **0.00** | 22.16, 14.21 | 20.12, 20.12 |
|  | 50% | 13.80, 13.80 | 40.50, 38.50 | 1.13, 0.84 | 41.53, 28.84 | 37.53, 34.25 |
|  | 75% | 20.33, 18.62 | 45.00, 44.25 | 4.28, 2.94 | 60.31, 42.30 | 51.65, 51.65 |
|  | 90% | 29.43, 24.28 | 60.60, 60.60 | 11.93, 11.34 | 65.12, 65.12 | 64.66, 64.66 |
| HEBO | 25% | 2.33, 1.54 | 16.00, 16.00 | **0.00**, **0.00** | 25.98, 18.04 | 10.07, 5.25 |
|  | 50% | 7.97, 6.04 | 23.00, 22.00 | 2.60, 0.85 | 52.46, 43.98 | 22.56, 21.52 |
|  | 75% | 14.31, 11.36 | 39.00, 37.50 | 4.80, 3.69 | 64.06, 52.93 | 39.62, 34.04 |
|  | 90% | 18.69, 16.99 | 45.00, 45.00 | 9.44, 9.44 | 72.56, 65.57 | 46.62, 42.54 |