# OpenReview forum: "Reinforced Few-Shot Acquisition Function Learning for Bayesian Optimization"
_NeurIPS.cc/2021/Conference — NeurIPS 2021 Poster_

### Official Review · Reviewer_9aK8 · 2021-07-12

**Rating:** 6
**Confidence:** 4

**Summary:**

This paper proposes to learn a general quickly-adaptable acquisition function (AF) using reinforcement learning. Particularly, it first connects AFs in BO with Q functions in RL, and then learns an adaptable AF with DQN + MAML. In order to mitigate the overfitting problem, it considers a Bayesian approach and applies Bayesian MAML. The empirical performances on a variety of black-box functions show very promising performance compared to many baseline BO algorithms.

Exploring the connection between AFs in BO and Q functions in RL is very interesting and novel AFAIK. Presenting BO as a RL problem allows one to learn a non-myopic AF which is also a good property compared to standard myopic AFs. Besides, training a fast adaptable AF at the presence of meta-data (tuning data of similar functions) is a nice contribution although there has been some prior work on learning an adaptable GP model.

The extensive experiments and ablation are useful and convincing.

My main concern of the paper that prevents me from giving a higher rating is on the use and the required amount of meta-data. Please see my question in below.

**Limitations And Societal Impact:**

The dependence on the amount of meta-data could be future discussed and ablated.

**Main Review:**

This paper makes a nice observation that AFs can be viewed as the Q function in RL. Then learning a non-myopic AF can be done with DQN. It would be nice to train a general AF that compares comparably or even better than standard BO methods. Unfortunately, this paper shows it is not the case at the end of experiments. (It's actually not clear to me how FSAF w/o update compares to other baselines, and I think this will be very useful to know) Therefore, the authors consider a few-shot setting, where the learned AF can be quickly adapted to the function of interest if given some meta data. They propose to meta-learn the base AF with MAML, but it overfits to the training data. So Bayesian MAML is further employed to mitigate the overfitting and stabilising the training.

The proposed model and the use of training algorithm method are technically	sound.

The entire model and training algorithm is complicated with multiple components, including GP, prior policy, Bayesian MAML, meta data, two replay buffers, etc. This paper makes a fairly good effort to explain the key components with the limited space. Nonetheless, it is not very straightforward to understand the entire algorithm. I would suggest moving Figure 6 to the main text for better clarity. A few key information is still missing from the main text, listed as follows:
- How do you define a stochastic policy given the Q function, and does this method work in a continuous state space?
- How do you generate the transitions in the reply buffer during meta-training?
- Where does the meta data come from and how do you construct the replay buffer of the meta data during both meta-training and evaluation?

It would be nice explain them in the revision.

My main concerns/questions about the paper are as follows,

- What is the meta-data used in adaptation. What does 1 shot mean? During meta-training, for every pair of support function(s) and target function, are they the same function? E.g., in the Electrical Grid Stability Dataset, the 60000 parameter configurations are divided into 40 sets from which 1 validation set is used for adaptation. Does it mean all the configurations are the input points of a shared black-box function and we have to evaluate all the configurations in the validation set to provide the meta data (totally 1500 configurations)? If so, it becomes meaningless to look at the optimisation performance with additional dozens of samples.

- In evaluation, is meta-data used by other BO methods e.g. EI, PI, GP-UCB? I guess it's straight ward to fit the GP hyper-parameters with the meta data, and if the meta data comes from the same function to optimise, we can even get the posterior to start sampling.

- Related to the question about the definition of policy defined from Q. If \pi_\theta is defined as softmax over the Q output logits, there is a normalisation term implicitly in Eq. 5. That term will become a problem in evaluating the prior in Eq 6.

- The prior distribution in Eq 6 could be un-normalizable if \delta doesn't decay fast enough as \theta moves to infinity.

- How does FSAF perform in the region with more samples than 80? A typical BO algorithm usually run from 10s to 100s. Does the discounting factor impact how long FSAF can generalise for? Can you please have some comments on the choice of reward function g(.)?

- How does FSAF scale to higher dimensional input space?


Some minor comments:

- Related work on meta BO: Chen et al, (2017) applies meta-learning to train a neural black-box optimization algorithm directly, instead of learning the AF, using GP samples as training functions.

Line 322 in p8: Figure 4 --> Figure 3. Figure 4 should be referred to from the paragraph of line 328.


Refs:

Chen, Y., Hoffman, M.W., Colmenarejo, S.G., Denil, M., Lillicrap, T.P., Botvinick, M. and Freitas, N., 2017, July. Learning to learn without gradient descent by gradient descent. In International Conference on Machine Learning (pp. 748-756). PMLR.


**Time Spent Reviewing:**

3

---

> ### Author Response · Authors · 2021-08-10
> **Response for Reviewer 9aK8**
>
> We greatly appreciate the reviewer for the overall positive feedback and the detailed suggestions for improving our paper. We provide our point-by-point response as follows:
>
> (D1) How do you define a stochastic policy given the Q function, and does this method work in a continuous state space?
>
> To obtain a stochastic policy, we take the common approach in DQN and use a softmax operation with the Q-values being the logits, i.e., $\pi(a|s)\propto \exp(Q(s,a))$. Recall that in FSAF, a Q-network has the joint state-action representation $(s,a)$ as its input, and we can get the corresponding Q-value $Q(s,a)$ at the output of the Q-network by a forward pass. Therefore, this design would work in both discrete and continuous state spaces.
>
> (D2) How do you generate the transitions in the replay buffer during meta-training?
>
> Recall that in FSAF, there are two replay buffers: a Q-replay buffer $\mathcal{R}_Q$ for DQN and a demo replay buffer $\mathcal{R}_D$ for the demo policy. During each iteration of meta-training, new trajectories of transitions are collected under the policy induced by each DQN particle and then stored in the Q-replay buffer, and the transitions for the demo replay buffer are collected under the demo policy (e.g., EI). The data generation procedure is also summarized in Appendix C.
>
> (D3) Explain how the metadata is used by other BO methods.
>
> As in the conventional BO algorithms, we use the metadata to fit the GP hyperparameters (e.g., lengthscales) used by all the AFs. On the other hand, we do not use the metadata as the initial data points for getting the posterior distribution in testing since the metadata and the testing data come from *different* functions, as described by the HPOBench example in the response (All-2).
>
> (D4) Regarding the definition of policy defined from Q and that the prior distribution in Eq. (6) could be un-normalizable
>
> In the KL-regularized minimization framework, it appears quite common to use *improper* prior distributions. As described by [23], one popular example is to use a uniform prior distribution (i.e., $q_0(\theta) \propto 1$), which does not have a finite integral and hence is improper and un-normalizable. During the training process, all we need is $\delta(\pi_\theta,\pi_D)$ for calculating the gradient as shown in Eq. (7), and this term is approximated by $\hat{\delta}(\pi_\theta,\pi_D)=\frac{1}{|D’|}\sum_{(s,a)\in  D’}\log (\pi_{\theta}(s,a))$ defined in Line 202, with the help of a replay buffer.
>
> Reference: [23] Yang Liu, Prajit Ramachandran, Qiang Liu, and Jian Peng. "Stein Variational Policy Gradient." UAI 2017
>
> (D5) How does FSAF perform in the region with more samples than 80?
>
> Based on the experimental results in the response (All-1), FSAF still performs well with more samples than 80.
>
> (D6) The choice of reward function $g(\cdot)$
>
> As the goal of BO is to achieve low simple regret, it is reasonable to let the reward function $g(\cdot)$ be a strictly decreasing function of simple regret. Through experiments we found that $g(z)=-\log z$ ($z$ is simple regret) achieves the best overall performance. We hypothesize that this is because $g(z)=-\log z$ could better encourage FSAF to find near-optimal sample points than other reward choices (e.g., linear reward $g(z)=-z$). Using nonlinear reward functions also appears quite common in other goal-finding RL problems, such as Pendulum.
>
> (D7) How does FSAF scale to higher dimensional input space?
>
> Recall that FSAF uses a 4-tuple $(\mu_t(x), \sigma_t(x), y_t^*, \frac{t}{T})$ as the state-action representation. Due to this design, FSAF is agnostic to the input dimensionality and can directly scale to higher input dimension. Moreover, FSAF can also be combined with the additive GP framework proposed by [Kandasamy et al., 2015] (and later used by MES [12]) to handle higher-dimensional input spaces.
>
> Reference:
>
> [Kandasamy et al., 2015] Kirthevasan Kandasamy, Jeff Schneider, and Barnabás Póczos. "High Dimensional Bayesian Optimisation and Bandits via Additive Models." ICML 2015.
>
> (D8) Related work on meta BO: [Chen et al., 2017] applies meta-learning to train a neural black-box optimization algorithm directly, instead of learning the AF, using GP samples as training functions.
>
> Thank you for pointing this out. Despite that [Chen et al., 2017] does not address fast adaptation with metadata, it shares the application scope of optimizing general black-box functions via meta-learning with FSAF, as also pointed out by Reviewer doq2. We will cite [Chen et al., 2017] and emphasize this in the final version.
>
> (D9) Move Figure 6 to the main text for better clarity
>
> Thank you for the suggestion. We will move Figure 6 to the main text in the final version (given the additional page of the camera-ready version).

---

> > ### Comment · Reviewer_9aK8 · 2021-08-18
> > **Additional questions**
> >
> > Thanks a lot for the detailed responses. I have two follow-up questions.
> >
> > 1. In the answer of "(All-2) Regarding the amount of metadata", for all the other experiments than HPOBench, do all the parameter sets come from the same black-box function? E.g., in the Electrical Grid Stability Dataset, the 60000 parameter configurations are split into 39 testing sets and 1 validation set. Are all the 40 functions different or just the same function evaluated in different input points?
> >
> > 2. In a continuous state space, I suspect the current Q-value based method won't work because it will become very slow to sample an action from the softmax distribution defined as exp(Q(s, .))/Z. Also, Eq 5 requires to compute the log-density of action with an unknown normalization term depending on theta. Could you please confirm that?

---

> > > ### Author Response · Authors · 2021-08-24
> > > **Response to additional questions**
> > >
> > > We thank the reviewer for the follow-up questions and would also like to apologize for the delayed response (somehow we did not get the email notification from the system and just saw the update earlier today). We first provide our response to the second question as follows, and we will provide a detailed response to the first question shortly.
> > >
> > > - Regarding continuous action space: We agree with the reviewer that handling a continuous action space is one known inherent challenge of Q-learning-based algorithms. To address this under FSAF, one possibility is to construct a Sobol grid (as typically done by prior works on BO, such as MetaBO [38] and Spearmint [1]) for approximately evaluating the normalization term as well as the softmax distribution. Another possibility is to adapt the sampling technique of Amortized Q-learning [Tom Van de Wiele et al., 2020], which selects actions with the help of sampling from some parameterized proposal distribution.
> > > We will add one remark in Section 3 to highlight this in the final version.
> > >
> > > Reference:
> > >
> > > [Tom Van de Wiele et al., 2020] Tom Van de Wiele, David Warde-Farley, Andriy Mnih, Volodymyr Mnih, “Q-Learning in enormous action spaces via amortized approximate maximization,” 2020 (available at https://arxiv.org/abs/2001.08116)

---

> > > > ### Comment · Reviewer_9aK8 · 2021-08-31
> > > > **Would be nice to clarify this in the revision**
> > > >
> > > > Thanks for your answers. Applying a pure Q-learning based algorithm in a continuous action space without learning a policy at the same time is a difficult problem. I'm glad the authors acknowledge this fact. My question on the continuous action space is not a criticism for this particular paper but more for clarification because this limitation was not explicit discussed in the original paper, and the reviewer's original answer ("Therefore, this design would work in both discrete and continuous state spaces") makes it even more confusing to me.

---

> > > > > ### Author Response · Authors · 2021-09-01
> > > > > **Thank you for your feedback**
> > > > >
> > > > > We truly appreciate your valuable feedback and thank you for taking the time to participate in the discussion. We will incorporate your suggestions into the final version of the paper.

---

> > > ### Author Response · Authors · 2021-08-26
> > > **Response to additional questions -- part 2**
> > >
> > >
> > > We thank the reviewer for the follow-up questions. We provide our response to the first question as follows:
> > >
> > > > In the answer of "(All-2) Regarding the amount of metadata", for all the other experiments than HPOBench, do all the parameter sets come from the same black-box function? E.g., in the Electrical Grid Stability Dataset, the 60000 parameter configurations are split into 39 testing sets and 1 validation set. Are all the 40 functions different or just the same function evaluated in different input points?
> > >
> > > Recall that FSAF is meant to leverage the metadata to better adapt to various black-box functions during testing, regardless of whether the metadata and the testing functions originally belong to the same function or not. In the experiments, we focus on the more *general* setting where the metadata and the testing data have similar characteristics but are *not* assumed to originally belong to the same function. As a result, we did not use the metadata as the initial points for calculating the posterior mean and variance of the GP surrogate model. For example, under the Electrical Grid Stability Dataset, we generate 40 distinct black-box functions by splitting this real dataset and then take 1 out of 40 as metadata only for learning the GP hyperparameters and fast adaptation of FSAF. This experimental setup is aligned with the problem setting described above.
> > >
> > > On the other hand, given the sufficient information provided by the Electrical Grid Stability Dataset, it also appears feasible to alternatively interpret this dataset as a whole as a single black-box function, as suggested by the reviewer. From this perspective, one could consider yet a *different* setting where the metadata is used for both estimating the GP hyperparameters and providing initial points for GP posterior inference. To further study this, we further test the AFs under the Electrical Grid Stability Dataset by using the metadata as initial points (and the amount of metadata is the same as that used by FSAF). Similar to our response (All-1), we report the mean regrets at $t=80$ and $t=140$ as follows:
> > >
> > > |             mean 80|      FSAF|FSAF with initial points|    MetaBO|MetaBO with initial points|  MetaBO-T|MetaBO-T with initial points|        EI|EI with initial points|        PI|PI with initial points|       MES|MES with initial points|    GP-UCB|GP-UCB with initial points|  TAF-ME
> > > | --------- | -------- | --------- | --------- | --------- | --------- | --------- | -------- | --------- | --------- | -------- | -------- | -------- | -------- | -------- | -------- |
> > > |Grid Stability 12D|**4.517234**|  5.371667|  6.780565|  5.493489|  6.901031|  5.539509|  5.565548|  5.402807|  5.567666|*4.646392|  6.832581|  6.246267|  7.714322|  6.077341| 14.265330|
> > >
> > > |             mean 140|      FSAF|FSAF with initial points|    MetaBO|MetaBO with initial points|  MetaBO-T|MetaBO-T with initial points|        EI|EI with initial points|        PI|PI with initial points|       MES|MES with initial points|    GP-UCB|GP-UCB with initial points| TAF-ME|
> > > | --------- | -------- | --------- | --------- | --------- | --------- | --------- | --------- | --------- | -------- | -------- | -------- | -------- | -------- | -------- | -------- |
> > > |Grid Stability 12D|*3.138378|**2.717777**|  3.688224|  3.484455|  4.658438|  3.697668|  3.677804|  4.760484|  3.960223|  3.357508|  5.414393|  4.756096|  4.779127|  4.547271| 11.831652|
> > >
> > > (Note that by design TAF always requires metadata for constructing a mixture of experts, and the TAF results provided in (All-1) already involve the use of metadata as initial points. To ensure a fair comparison, we retest TAF by using the same initial points as the other AFs)
> > >
> > > We can see that adding initial points for obtaining the GP posterior appears helpful to the AFs in general, and FSAF still achieves comparable or better regrets than the benchmark methods.
> > > We also observe that adding initial points is not always helpful, and we conjecture that this is because the real dataset is likely to consist of rather noisy evaluations which do not necessarily provide enough information to their neighboring points.

---

> > > > ### Comment · Reviewer_9aK8 · 2021-08-31
> > > > **These results are very useful but deserve more analysis**
> > > >
> > > > Thanks for the additional experiment results. It is nice to see that FSAF still remains its advantage after including the meta data as initial data points. But I think the experiment results deserve more discussion (probably in the appendix of a revision). Take the Electrical Grid Stability Dataset for example, it has totally 60000 parameter configurations (data points). Each subset of a 40-splits consists of 1500 data points. So a BO method (e.g. PI) with initial points at step 0 is effectively a standard BO method at 1500 steps, and step 80 and 140 correspond to step 1580 and 1640. I'm actually *very surprised* to see that the results of a few BO method (e.g. EI) with initial points (effectively step 1580 or 1640) is comparable or even worse than EI without initial points. Moreover, the results of a BO method with initial points at step 80 or 140 shouldn't really have any statistically significant difference, but I see quite a big difference in all the BO baselines, e.g. comparing MetaBO-T with initial points at step 80 and 140!
> > > >
> > > > As a related comment, while we may ignore the fact that all the 39 test functions belong to the same black-box function as the meta-data function and compare them in the original setting with other BO methods, the fact that they are indeed part of the same function would mean the meta-data is more informative for transfer learning than the basic assumption that the test function have similar characteristics to the meta-data. A very important part of the method, how similar the test function is to the meta data / support function is not well discussed in the current version.

---

### Official Review · Reviewer_wSpJ · 2021-07-12

**Rating:** 7
**Confidence:** 5

**Summary:**

The authors propose to combine meta-learning, reinforcement learning and few-shot learning to learn an acquisition function for Bayesian optimization which is outperforming the hand-crafted alternatives. The proposed method is evaluated on a variety of problems against the most common acquisition functions.

**Limitations And Societal Impact:**

-

**Main Review:**

The authors nicely motivate and explain their method. The proposed method is in most places well-explained. There are few cases where the additionally information is hidden in the appendix. The components of the method and the decisions made by the authors are well explained and supported by ablation studies whenever needed. The experimental section provides strong evidence that the proposed method is an improvement over the state-of-the-art.

I think the authors could highlight the fact that FSAF is trained on synthetic data (which we get for free) and requires only metadata for very few tasks (which might be costly to generate). I point that out because typically meta-learning methods use many tasks and infer from them to a new task so the reader might just expect the very same setup. I believe this is a strong point of the method and I only realized this after actively looking for which metadata is used in the appendix.

The authors claim that the proposed method requires only few tasks and therefore requires less metadata than related approaches such as [16]. One question unanswered is how many observations per task are required. In practice, it is probably more common to have metadata for many tasks but for each only few observations. I hope the authors can be a bit more specific under which circumstances their method works and under which it doesn’t.

I am curious how this method compares to [16] and the acquisition function used by the winners of the NeurIPS BBO challenge: https://arxiv.org/abs/2012.03826

Why was the median instead of the mean reported? This is rather uncommon and might indicate that the method has some problems learning good solutions every time.

Line 289: N, K and S are not used in the main paper. A short remark pointing to the appendix would be helpful here.

**Time Spent Reviewing:**

6

---

> ### Author Response · Authors · 2021-08-10
> **Response for Reviewer wSpJ**
>
> We sincerely thank the reviewer for the positive feedback and the helpful suggestions. We provide our point-by-point response as follows.
>
> (C1) Highlight the fact that FSAF is trained only on synthetic data
>
> Thank you for pointing this out! Indeed, FSAF is trained solely on synthetic GP functions (with training details provided in Appendix A) and is able to adapt to a broad variety of black-box functions based only on a small amount of metadata. We will further highlight this strength in the final version.
>
> (C2) Regarding the median regrets in the experimental results
>
> As MetaBO [38] serves as one critical baseline in meta-learning for BO and achieves superior performance in terms of *median regret* as shown by [38], we chose to use the same performance metric as [38] and report the median as well as other percentiles in the submitted manuscript.
> Based on the table provided in the response (All-1), we can see that FSAF still achieves comparable or better regrets than the benchmark methods in all the tasks in terms of *mean regret*.
>
> (C3) N, K and S are not used in the main paper
>
> Thank you for catching this. We will add a remark pointing to Algorithm 1 in Appendix C to address this in the final version.

---

### Official Review · Reviewer_doq2 · 2021-07-16

**Rating:** 7
**Confidence:** 4

**Summary:**

The paper proposes "Reinforced Few-Shot Acquisition Function Learning" (FSAF), a meta-learning approach for designing few-shot acquisition functions (AFs) for Bayesian Optimization (BO).

The authors map BO onto an reinforcement learning (RL) problem, i.e., identify the AF with an RL policy which proposes evaluation points, given the current state of the optimization in form of the Gaussian Process (GP) posterior (and further features). The authors develop a Bayesian version of Deep Q-learning (DQN), leveraging Stein Variational Gradient Descent (SVGD), and combine it with ideas from Bayesian MAML (BMAML) to meta-train an initial AF on function samples from GP priors. This initial AF can be adapted to a family of target optimization problems using a few (~1-5) context optimization problems from this family for fast (few-shot) adaptation.

The method is evaluated on various sets of optimization problems, showing favorable performance compared to existing hand-designed AFs (EI, PI, GP-UCB, MES) as well as the transfer-learning approach MetaBO [1].

**Ethical Concerns:**

No ethical concerns.

**Limitations And Societal Impact:**

No potential negative societal impact.

**Main Review:**

Originality: FSAF is a novel meta-learning approach for AF-design. It stands out from existing work in that it focuses on few-shot adaptation, i.e., the architecture is trained to use a small context set of unseen optimization problems to quickly adapt to a given class of optimization problems, which is a novel approach to meta-learning for BO.

While the authors adequately describe that their proposed algorithm is a novel combination of existing work (DQN, SVGD, BMAML), I have several concerns about the presentation of the general setup and the application scopes of the compared methods:

- The authors do not state clearly that the general idea for mapping BO onto an RL problem (Sec. 3.1) is not new: [1] propose to use exactly the same general RL setting with the same state-action representation and reward function (indeed, the frameworks are similar enough that the authors can just extend the software framework from [1] by their novel training algorithm). This is not adequately cited.
- Similarly, I feel that the presented experimental comparison to [1] can be misleading. [1] is a its core transfer-learning algorithm, i.e., belongs to the class of algorithms discussed in ll. 39-51 (where [1] is not discussed). Indeed, [1] devise an algorithm which is supposed to be trained on optimization problems which come from the same distribution as the target problems and do not consider a fast adaptation step. In contrast, the authors explicitly train their AF for fast adaptation. I appreciate that the authors consider the dimensionality-agnostic MetaBO [1] version trained on GP samples and also test a naively fine-tuned version (MetaBO-T) and state that [1] does not consider a fast adaptation step, but still I feel that the comparison to [1] is not adequate, as [1] was conceived with a different application scope in mind (transfer learning vs. few-shot adaptation). If the incentive of the authors is to compare to transfer-learning methods, they should also include comparisons to, e.g., ensemble methods for transfer learning as discussed in ll. 39-51 (e.g., the TAF framework proposed in [2]) and use the context set of $K$ optimization problems as the source tasks.
- The authors do not cite [3], which propose learning global optimizers in a very similar setting as FSAF. Indeed, in contrast to the aforementioned class of transfer-learning methods [1,2], [3] train a general-purpose optimizer (much like FSAF) on GP samples and evaluate the resulting optimizer on a range of different target distributions. While [3] also do not consider a fast adaptation step, this work fits the setting of FSAF better than [1,2]. It is known that the results of [3] are hard to reproduce, so I do not expect the authors to provide experimental comparisons, but I feel that the experimental evaluation lacks a clear discussion of the application scopes of the compared methods.
- As noted in the previous paragraph, I feel that natural competitors for FSAF are general-purpose global optimizers (instead of approaches like [1,2] which are designed specifically for the transfer-learning setup). I appreciate the comparisons to general-purpose AFs (EI, PI, MES) but I would also like to see comparisons as presented in [3], e.g., to a highly optimized BO framework like Spearmint [4].

Quality: The author's approach to design a few-shot AF for BO is interesting and the proposed combination of DQN, SVGD, and BMAML is clever and appropriate.

Clarity: The overall writing and organization of the paper is good. Please refer to my notes under "Originality" regarding my concerns about the clarity of the experimental evaluation.

Significance: The proposed method is an interesting and important contribution, as — to the best of my knowledge — it is the first approach for few-shot AF learning in the context of BO. Nevertheless, I feel the paper is not ready for publication yet due to my concerns regarding the experimental evaluation and the lack of discussion regarding the application scopes of the compared methods. I therefore lean to rejection but am I willing to increase my score if the authors address my concerns.

References:

[1] Volpp et al., "Meta-Learning Acquisition Functions for Transfer Learning in Bayesian Optimization", ICLR 2020

[2] Wistuba et al., "Scalable Gaussian-process Transfer Surrogates for Hyperparameter Optimization", JMLR 2018

[3] Chen et al., "Learning to Learn without Gradient Descent by Gradient Descent", ICML 2017

[4] Snoek et al., "Practical Bayesian Optimization of Machine Learning Algorithms", NeurIPS 2012


-------------------------------------------------------------------------
Update after reading the author's response:

I thank the authors for their response. I raise my score because the authors

- clarify the application scope of FSAF and the baseline methods ("general-purpose optimizer" vs. "transfer-learned optimizer" vs. "meta-learned/fast adaptation optimizer")
- include a further transfer-learning approach (TAF) in their comparison
- include general-purpose optimizers (Spearmint, HEBO) in their comparison
- improve the citation of related work (Chen et al., Volpp et al.)

**Time Spent Reviewing:**

6

---

> ### Author Response · Authors · 2021-08-10
> **Response for Reviewer doq2**
>
> We greatly appreciate the reviewer’s constructive feedback for improving our paper. We provide our point-by-point response as follows.
>
> (B1) Regarding the general idea for mapping BO onto an RL problem and the connection between FSAF and MetaBO
>
> We agree with the reviewer that FSAF is not the first approach that addresses BO through the lens of RL, and the main contribution of this paper is to present the first few-shot learning based acquisition function for BO. In the submitted manuscript, we have mentioned that MetaBO is an existing well-performing RL-based solution to BO. For example:
> - In Line 281, we stated that “MetaBO is a neural AF trained via policy-based RL and recently achieves superior performance in BO.”
> - In Line 374, we stated that “MetaBO [38] used policy-based RL to train a neural AF that learns structural properties of a set of source tasks to enable knowledge transfer to related new tasks.”
>
> We will make this fact more clear in the final version.
>
> Regarding the state-action representation, FSAF uses a 4-tuple $(\mu_t(x), \sigma_t(x), y_t^*, \frac{t}{T})$, and MetaBO uses $(\mu_t(x), \sigma_t(x), x, t, T)$ (cf. Section 4 of [38]). As both FSAF and MetaBO address BO through the lens of RL, they share some common choices for the representation (e.g., $\mu_t(x)$ and $\sigma_t(x)$). Despite the similarity in the representation and the fact that using a good state-action representation is important in RL, we found that a good representation itself does not guarantee good regret performance. As shown by the results in Figure 1, DQN+MAML uses the same state-action representation as FSAF, but it still suffers from severe overfitting and poor regret. Accordingly, in this paper, we show that the key to achieving low regret under various black-box functions is the appropriate use of a small amount of metadata through the design of few-shot adaptation in FSAF.
>
> Based on the above discussion, we will add one remark in Section 3.1 to further highlight the above connection and comparison between FSAF and MetaBO in the final version.
>
> (B2) The compared benchmark methods and their application scopes
>
> We agree with the reviewer that MetaBO is not designed for few-shot fast adaptation but for transfer learning in BO. In contrast, FSAF is positioned as a meta-learning BO algorithm that can leverage a small amount of metadata for few-shot fast adaptation. Despite the difference, we choose MetaBO and its fine-tuned version MetaBO-T as important benchmark methods since there are not many meta-learning BO algorithms and MetaBO appears to be a strong benchmark method in the class of meta-learning for BO, as shown in [38]. To make a more thorough comparison, we have provided the experimental results of TAF in our response (All-1).
>
> In addition, we thank the reviewer for suggesting a comparison between FSAF and the general-purpose global optimizers (such as Spearmint [1] and [Chen et al., 2017]) to further demonstrate the capability of FSAF. While Spearmint and [Chen et al., 2017] do not address fast adaptation with metadata, they do share the application scope of optimizing general black-box functions with FSAF. We will cite [Chen et al., 2017] and emphasize this angle in the final version. The experimental results of Spearmint are also provided in the table in our response (All-1).

---

### Official Review · Reviewer_dJ7J · 2021-07-16

**Rating:** 7
**Confidence:** 2

**Summary:**

This paper proposed an algorithm for few-shot Bayesian optimization. It conducted experiments to show that their algorithm could adapt to a variety of black-box function and prevent overfitting.

**Limitations And Societal Impact:**

This author adequately addressed the limitations in this paper. It is up front about the similarity between their algorithm and former work in remark and  related work. Their work does not have potential negative societal impact. It also addressed its need for meta-data in conclusion.

**Main Review:**

Originality. Although the algorithm in this paper might be similar with some former work, it applied some technique to improved both its validity and novelty. It also cited related work clearly.

Quality. This is a complete work. Their experiment result showed that the algorithm they design is valid. However, it used median of regret instead of mean, which might require further explanation.

Clarity. This paper is well-written. However, some details might need further explanation. For example, in line 136, it said "Based on the conceptual similarity between acquisition functions and Q-functions,.....". However, the similarity is not that clear, since it seems that there are not any concepts in Bayesian optimization that are similar with transition in RL. Maybe it is more similar with the reward in bandits. Besides, 'Section 3.2' in line 159 might be 'Section 3.1'.

Significance. The algorithm in this paper can be applied to general black-box function, which might be useful in application and allow people to build better algorithm in Bayesian optimization based on their algorithm.

**Time Spent Reviewing:**

8

---

> ### Author Response · Authors · 2021-08-10
> **Response for Reviewer dJ7J**
>
> We greatly thank the reviewer for the positive feedback and the insightful suggestions. We provide our point-by-point response as follows.
>
> (A1) Regarding the median regrets in the experimental results:
>
> As MetaBO [38] serves as one critical baseline in meta-learning for BO and achieves superior performance in terms of *median regret* as shown by [38], we chose to use the same performance metric as [38] and report the median as well as other percentiles in the submitted manuscript. In the response (All-1), we report the *mean regrets* of FSAF as well as the benchmark methods to demonstrate that FSAF also achieves superior regret in the mean.
>
> (A2) Regarding the conceptual similarity between acquisition functions and Q-functions:
>
>
>
> BO is a sequential decision making problem, and an acquisition function (AF), denoted by $\Psi(x;F_t)$ in Section 2 of the submitted manuscript (with $F_t$ denoting the observations up to step $t$), provides a way to express low-complexity index-based policies, under which the next query is determined by $x_{t+1}=\arg\max_{x} \Psi(x;F_t)$. In the context of GP, the information about $x$ and $F_t$ is usually encoded by the posterior mean $\mu_t(x)$ and posterior variance $\sigma^2_t(x)$. The evolution from {$(\mu_t(x),\sigma^2_t(x))$} to {$(\mu_{t+1}(x),\sigma^2_{t+1}(x))$} can be viewed as a “transition” triggered by the action (i.e., selecting $x_{t+1}$ as the query point) from the perspective of RL.
>
> On the other hand, to address sequential decision making via RL, a Q-learning-based RL algorithm (e.g., DQN) would learn a Q-function $Q(s,a)$ and choose the actions by $\arg\max_a Q(s,a)$. From this perspective, the learned Q-function bears functional similarity to AFs. By properly designing a state-action representation and reward signals that reflect the goal of BO (i.e., achieving low regret), we can train a Q-function that plays the role of a non-myopic AF for BO.

---

### Author Response · Authors · 2021-08-10
**Response for All Reviewers**

We greatly thank all the reviewers for their valuable feedback and insightful suggestions! We are glad to see that the reviewers appreciate our contributions in presenting the first few-shot acquisition function for BO.

We first provide our response to those questions asked by multiple reviewers in this thread, and we post our responses to the rest of the questions below the review comments of each individual reviewer.

---

(All-1) Compare FSAF with additional benchmark methods: TAF, Spearmint, and HEBO (the winner of BBO challenge in NeurIPS 2020)

We show the regret performance of FSAF as well as the benchmark methods in the table below. For TAF, we use a mixture of five experts and the same metadata as FSAF. For Spearmint, we use the source code and the default setting provided by the authors of [1] at https://github.com/JasperSnoek/spearmint. For HEBO, we use the open-source implementation and the default setting provided by the authors of HEBO at https://github.com/huawei-noah/noah-research/tree/master/BO/HEBO. We report the *mean regrets* measured at $t=80$ and a larger $t=140$, as suggested by the reviewers. All the values displayed in the table are scaled by $10^2$ (e.g., 5 means that the regret is 0.05) for better readability. Regarding Spearmint and HEBO, due to their extremely high computation times, we try our best to finish the experiments for 5 tasks (3 real datasets and 2 optimization benchmark functions). For better readability, the best and the second best methods for each task are highlighted in bold and with an asterisk, respectively.

|             mean 80|      FSAF|    MetaBO|  MetaBO-T|        EI|        PI|       MES|    GP-UCB|    TAF-ME| Spearmint|      HEBO|
| --------- | -------- | --------- | --------- | --------- | --------- | --------- | --------- | --------- | -------- | -------- |
|        Eggholder 2D|*7.349308| 15.116352| 13.358023| 11.028137| 11.683494| 12.298864| 13.097762| 11.804124|**7.093724**| 14.946570|
|           Ackley 2D|**5.161302**| 22.890568| 15.371699| 10.033104|*9.744870| 14.893383| 17.729878|  9.904663|       -|       -|
|  Styblinski-Tang 2D|**1.508214**|  1.509041|  1.508522|  1.515484|  3.748981|  1.659405|*1.508325|  1.526830|       -|       -|
|      Dixon-Price 7D|*0.229449|  0.444428|  0.386581|**0.205350**|  0.570960|  2.554704|  4.313532|  0.255002|       -|       -|
|          Powell 10D|  2.636913|  8.951325|  8.883294|*2.382136|  5.462201|  9.251378|  4.053106|  3.345595|  4.130770|**1.989607**|
|     HPOBench XGB 6D|**2.336687**|*2.338523|  3.520234|  2.482445|  2.398790|  2.391101|  2.391029|  6.285634|  5.036511|  3.366748|
|  Grid Stability 12D|**4.517234**|  6.780565|  6.901031|*5.565548|  5.567666|  6.832581|  7.714322| 14.813416| 13.847793|  7.966677|
|           PM2.5 14D|**10.241379**| 22.689655| 24.655172| 17.103448| 14.517241| 17.689655| 18.793103|*11.206897| 38.350000| 26.384615|
|              Oil 4D| 10.206897|  9.581379|**5.799310**|*9.147241|  9.387931|  9.596207| 10.520000| 13.236897|       -|       -|
|        Asteroid 12D|**0.000000**|  5.679093|*0.050281|**0.000000**|  0.147946|  0.717272|  0.440438|  7.547428|       -|       -|

|             mean 140|      FSAF|    MetaBO|  MetaBO-T|        EI|        PI|       MES|    GP-UCB|    TAF-ME| Spearmint|      HEBO|
| --------- | -------- | --------- | --------- | --------- | --------- | --------- | --------- | --------- | -------- | -------- |
|        Eggholder 2D|**4.565881**|  9.349542|  7.293597|  6.506231|  8.079134|  8.741064|  8.182243|*6.044781|  6.305555| 11.124406|
|           Ackley 2D|**5.161302**| 18.667136| 11.361969| 10.033104|  9.744870| 14.893383| 17.729878|*7.854078|       -|       -|
|  Styblinski-Tang 2D|**1.508208**|  1.508479|  1.508348|  1.512736|  3.388902|  1.630991|*1.508283|  1.523938|       -|       -|
|      Dixon-Price 7D|  0.229449|  0.423935|  0.347227|**0.200969**|  0.570960|  2.417500|  4.198874|*0.221225|       -|       -|
|          Powell 10D|*1.813249|  5.932667|  5.030146|  2.042510|  4.615098|  9.143156|**1.706708**|  2.427002|  3.982981|  1.820002|
|     HPOBench XGB 6D|**1.784274**|*1.816487|  2.091657|  1.935310|  2.149878|  2.038846|  2.213091|  6.285634|  5.002538|  2.506331|
|  Grid Stability 12D|**3.138378**|  3.688224|  4.658438|*3.677804|  3.960223|  5.414393|  4.779127| 14.729069| 13.231192|  7.247433|
|           PM2.5 14D|**7.517241**| 16.724138| 17.000000| 14.103448| 10.344828| 15.551724| 14.068966|*8.689655| 38.350000| 24.282051|
|              Oil 4D|  3.927586|  4.134138|*3.477931|  5.506897|**2.640690**|  3.928276|  4.586207|  9.316897|       -|       -|
|        Asteroid 12D|**0.000000**|**0.000000**|**0.000000**|**0.000000**|**0.000000**|*0.050281|**0.000000**|  5.977264|       -|       -|


- FSAF achieves overall comparable or better regrets than the benchmark methods (including the highly optimized benchmarks Spearmint and HEBO) in all the tasks.

- Spearmint performs well under the lower-dimensional Eggholder function but has high regrets in higher-dimensional problems (e.g., Grid Stability 12D and Powell 10D). This appears consistent with the results provided by [Chen et al., 2017].

- TAF suffers from high regrets in multiple real-world test functions (e.g., HPOBench 6D). Under optimization benchmark functions, TAF performs relatively well at $t=140$ but no so well at the smaller $t=80$.

- HEBO is strong in Powell 10D but suffers in Eggholder 2D and PM 2.5 14D. We conjecture that this is because HEBO uses the maximum of EI, PI, and GP-UCB as their AF, and these three AFs perform quite well in Powell but not well in Eggholder and PM 2.5.

Reference:

[Chen et al., 2017] Yutian Chen, Matthew W. Hoffman, Sergio Gómez Colmenarejo, Misha Denil, Timothy P. Lillicrap, Matt Botvinick, and Nando Freitas. "Learning to Learn without Gradient Descent by Gradient Descent", ICML 2017

---

(All-2) Regarding the amount of metadata

As this question has been raised by multiple reviewers, we think that it would be helpful to first clarify the definition of a "shot" for fast adaptation in this paper. Based on the convention of few-shot learning for RL, one *shot* in few-shot adaptation refers to a trajectory or a rollout generated under the target task. For example, in a MuJoCo robot control task, a *shot* would correspond to a complete trajectory of states and actions collected under a policy (cf. MAML [18] and BMAML [20]). Next, to better describe this in the context of BO, let us consider the HPOBench dataset for XGBoost Hyperparameter Optimization as an example (the details of the dataset can also be found in Appendix B.3):
- The HPOBench dataset provides the pre-computed results of XGBoost with six tunable hyperparameters (e.g., learning rate, L1 and L2 regularization terms, and subsampling ratios) on 48 classification datasets, each of which is associated with 1000 randomly selected hyperparameter configurations. Accordingly, these 48 subsets of pre-computed results can be naturally viewed as 48 different black-box test functions for BO (with the cardinality of the input domain = 1000, for each black-box function).
- While these 48 black-box functions are different, they are deemed to be of common characteristics as these functions are obtained under the same XGBoost algorithm and related classification datasets. Based on this, as typically done in BO or GP regression (e.g., in [16]), we use 1 (out of 48) black-box function as the validation dataset for learning the GP hyperparameters (e.g., lengthscale). The remaining 47 black-box functions are used only for testing.
-In addition to using this small validation dataset for learning the GP hyperparameters, we further leverage this dataset for fast adaptation in FSAF. Specifically, each Q-network would perform sequential sampling based on its Q-value for generating one trajectory of length = 50 steps and then use this trajectory for the gradient updates for fast adaptation. This choice corresponds to *one-shot* adaptation since only one trajectory is used for the fast adaptation of each Q-network (cf. BMAML [20, Appendix C.2]). Notably, the choice of 50 steps is reasonable as it is expected that at least a few tens of sample points are needed in order to properly identify the characteristics of a black-box function with a multi-dimensional input domain.

The use of metadata is similar in other tasks. As illustrated by the above HPOBench example, FSAF would be a promising approach to those global optimization problems with multiple black-box functions of similar characteristics and a small amount of metadata available (usually a few tens to a few hundreds of data points).

---

### Decision · Program_Chairs · 2021-09-27

**Decision:**

Accept (Poster)

**Comment:**

The paper proposed a novel approach that combines meta learning and reinforcement learning for designing few-shot acquisition functions for Bayesian Optimization. All reviewers find the problem setup interesting and appreciate the novelty and applicability of the proposed algorithm. After a few rounds of interaction during the discussion phase, the reviewers are convinced about the empirical significance of the proposed work.

When preparing a revision, the authors are strongly encouraged to take into account the reviews and accommodate the changes reflected in the author discussions---in particular, to further strengthen the empirical analysis, incorporate new references, clarification of the technical challenge, as well as elaborate on details of the FSAF algorithm and its application scope.